

# PPCon 1.0: Biogeochemical Argo Profile Prediction with 1D Convolutional Networks

Gloria Pietropolli[1, 2], Luca Manzoni[1, 2], and Gianpiero Cossarini[1]

[1]National Institute of Oceanography and Applied Geophysics - OGS, Trieste, Italy.
[2]Dipartimento di Matematica e Geoscienze, University of Trieste, Trieste, Italy.

**Correspondence:** Gloria Pietropolli (gloria.pietropolli@phd.units.it)

**Abstract.** Effective observation of the ocean is vital for studying and assessing the state and evolution of the marine ecosystem, and for evaluating the impact of human activities. However, obtaining comprehensive oceanic measurements across temporal and spatial scales and for different biogeochemical variables remains challenging. Autonomous oceanographic instruments, such as Biogeochemical (BCG) Argo profiling floats, have helped expand our ability to obtain subsurface and deep-ocean

measurements, but measuring biogeochemical variables such as nutrient concentration still remains more demanding and expensive than measuring physical variables. Therefore, developing methods to estimate marine biogeochemical variables from high-frequency measurements is very much needed. Current Neural Network (NN) models developed for this task are based on a Multilayer Perceptron (MLP) architecture, trained over punctual pairs of input-output features. However, MLPs lack awareness of the typical shape of biogeochemical variable profiles they aim to infer, resulting in irregularities such as jumps

and gaps when used for the prediction of vertical profiles. In this study, we evaluate the effectiveness of a one-dimensional Convolutional Neural Network (1D CNN) model to predict nutrient profiles, leveraging the typical shape of vertical profiles of a variable as a prior constraint during training. We will present a novel model named PPCon (Predict Profiles Convolutional), which is trained over a dataset containing BCG Argo float measurements, for the prediction of nitrate, chlorophyll and backscattering (bbp700), starting from the date, geolocation, temperature, salinity, and oxygen. The effectiveness of the model

is then accurately validated by presenting both quantitative metrics and visual representations of the predicted profiles. Our proposed approach proves capable of overcoming the limitations of MLPs, resulting in smooth and accurate profile predictions.

## 1   Introduction

Observation of the ocean is crucial for studying the state and evolution of the marine ecosystem and assessing the impact of human activities (Campbell et al. (2016); Euzen et al. (2017)). Access to reliable and extensive oceanic measurements remains

restricted due to the challenges of collecting comprehensive observations on multiple temporal and spatial scales, as well as variability in the availability of observations across different biogeochemical variables (Munk (2000)).

The introduction of autonomous oceanographic instruments such as Biogeochemical (BGC) Argo floats have notably expanded our ability to obtain subsurface and deep ocean measurements (Miloslavich et al. (2019)). BGC-Argo floats are autonomous profiling platforms that incorporate physical and biogeochemical sensors, enabling to collect time-series of vertical



profiles across various sea conditions and throughout the complete annual cycle (d'Ortenzio et al. (2014); Mignot et al. (2014)). Over the past decade, there has been a steady rise in the number of biogeochemical profiles acquired using these platforms (Johnson et al. (2013); Johnson and Claustre (2016)). These instruments are essential to advance our knowledge of the biogeochemical state of the ocean, as one of their principal advantages is the assimilation into ocean biogeochemical models (Mignot et al. (2019); D'ortenzio et al. (2020)). This assimilation process is particularly promising for variables such as oxygen, ni-

trate, and chlorophyll concentrations, as they serve as core state variables in most ocean biogeochemical models (Teruzzi et al. (2021); Cossarini et al. (2019)).

However, the measurement of biogeochemical variables such as nutrient concentration and carbonate system variables (e.g. nitrate, chlorophyll, and pH) remains more demanding and expensive compared to physical variables (e.g. temperature, salinity) and oxygen. In fact, among the BCG sensors, oxygen is the most commonly measured variable: there have been approximately

$250,000$ oxygen profiles collected worldwide, which is twice the number of profiles for chlorophyll, and more than four times the number of profiles for nitrate and bbp700 (https://biogeochemical-argo.org. Thus, developing methods to estimate low-frequency marine biogeochemical variables from high-frequency measurements is essential to maximize the potential of observing systems such as the Argo program. Major efforts have been devoted toward the improvement of the long-term reliability and accuracy of autonomous measurements in recent years (Sauzède et al. (2017)).

Artificial neural networks (ANNs) are computational models that are inspired by the structure and function of the human brain and have become a widely used approach for solving complex problems in a variety of fields, from computer vision and natural language processing to finance and engineering (Krogh (2008)). ANNs have emerged as a powerful tool for modeling complex, non-linear relationships also in the oceanographic field, where their use has seen a significant increase in recent years (Ahmad (2019)). The use of these models has found applications in a wide range of areas, such as oceanic climate prediction

and forecasting (Mori et al. (2017)), species identification (Goodwin et al. (2014)), coastal morphological and morphodynamic modeling (Goldstein et al. (2019)), ocean current prediction (Bolton and Zanna (2019)), interpolation and gap filling for remote sensing observation (Sammartino et al. (2020)), and the integration of observation data into biogeochemical models (Pietropolli et al. (2022)). These examples demonstrate the broad utility of ANNs in advancing our understanding of the ocean and its processes.

Existing ANN-based techniques to infer low-sampled variables starting from high-sampled ones are based on Multilayer Perceptron (MLP) architecture, a type of feedforward NN that processes input data through interconnected layers of nodes, or neurons, with each neuron in a layer receiving inputs from all the neurons in the previous layer (Taud and Mas (2018)). The initial model designed for this task was proposed by Sauzède et al. (2017), where a deterministic MLP network, named *CANYON*, was trained on a global ocean dataset to estimate biogeochemically relevant variables from concurrent in situ sam-

ples of temperature, salinity, pressure and oxygen and their latitude, longitude, depth, and date. Later, an improved version, called *canyon-b*, was introduced by Bittig et al. (2018). In this approach, a Bayesian framework was utilized, and experimental findings demonstrated that this method resulted in a more robust output. This methodology was subsequently limited to the Mediterranean Sea, resulting in the development of *canyon-med* by Fourrier et al. (2020), and empirical results validated the effectiveness of restricting the model to a smaller region. The latest advancement in this field is presented in Pietropolli et al.





(2023a), wherein the authors enhance the performance related to Mediterranean Sea predictions by leveraging a more extensive training dataset and implementing a two-step quality check procedure to improve its quality.

Despite their widespread use, applications based on MLPs currently lack awareness of the typical shape of biogeochemical variable profiles they aim to infer. In fact, when these methods are used to forecast profiles from Argo float measurements, they may generate jumps and irregularities in the reconstruction. This originates from the fact that MLPs are trained on individual

data points and provide pointwise outputs, which makes the generation of regular profiles challenging as the NN does not take into account the vertical neighbors of predicted variables.

To solve this problem effectively, our idea consists of working directly with an architecture that infers the complete vertical profile. This approach takes advantage of architectures like the Convolutional Neural Network (CNN) that operate on vector inputs instead of individual points. CNNs are recognized as one of the most impressive forms of ANNs, especially for their

effectiveness in tackling complex pattern recognition problems (O'Shea and Nash (2015); Gu et al. (2018)). While CNNs are well-known for their success in image classification tasks, they can also be used for other tasks such as speech recognition (Shan et al. (2018)), natural language processing (Collobert et al. (2011)), and even drug discovery (Goh et al. (2017)).

In this study, we evaluate the effectiveness of a one-dimensional (1D) CNN model (Kiranyaz et al. (2021)) for predicting nutrient vertical profiles from input data such as sampling time, geolocation, and profiles of temperature, salinity, and oxygen,

using Argo float measurements as the training dataset. This approach, called PPCon (Predict Profile Convolutional) is applied to generate synthetic profiles of nitrate, chlorophyll, and backscattering (bbp700). Thanks to the intrinsic spatial-aware nature of its CNN architecture, PPCon is able to leverage the typical shape of vertical profiles of a variable as a prior constraint during training. PPCon predictions are characterized by lower error with respect to the one obtained with MLP while also showing smoother predictions and the disappearing of phenomena such as gaps and irregularities in the generation of vertical profiles.

This paper is organized as follows: Section 2 presents the dataset utilized for training the deep learning (DL) architecture, including its key characteristics. Section 3 provides a detailed overview of the PPCon approach, encompassing the architecture, preprocessing techniques applied to input data, and the specialized loss function employed for network training. In Section 4, we outline the specific experimental settings employed to enable complete reproducibility of the PPCon architecture. Section 5 presents a summary of the key results obtained during the experimental campaign we conducted to validate our proposed

techniques, and finally, Section 7 presents the conclusions drawn from our work and directions for future research.

## 2  Dataset: the Argo GDACs

The data used to train and test the architecture discussed in this paper comes from the Array for Real-time Geostrophic Oceanography (Argo) program (Bittig et al. (2019)), specifically the Argo float collecting also biogeochemical variables (BCG Argo float). This program is an important part of the Global Ocean Observing System (GOOS) (https://www.goosocean.org/)

and is dedicated to monitoring changes in the temperature and salinity of the upper ocean. The Argo program was primarily designed to observe pressure, temperature, and salinity (conductivity) within the upper 2000 meters of the ocean. However, due to advancements in float and sensor technologies, newly developed sensors now enable profiling floats to accurately



observe biogeochemical properties. Over the past decade, there has been a consistent and substantial increase in the number of biogeochemical profiles obtained through BGC-Argo float platforms. For instance, by 2011, the global ocean had accumulated approximately $45,000$ BGC-Argo profiles across all parameters, while, by 2017, this number had risen to almost $390,000$ profiles. Thus, the BGC-Argo floats network has become a crucial component of the ocean observing system, enabling the monitoring, understanding, and prediction of changes in the ocean ecosystem. As of now, the Argo program has amassed over 2 million data profiles, and analysis of this data has made significant contributions to basic research as well as national and international climate studies (Jayne et al. (2017)).

Our investigation specifically centers on the dataset made available by the Argo Global Data Assembly Centers (GDACs) (e.g. Coriolis, NOAA, among others), which disseminate Delayed Mode (DM) data procured from 11 data centers situated across 9 countries. DM data undergoes a more exacting quality control process and is typically released a few months later to their sampling (Li et al. (2020)). GDACs also supply Real-Time Mode (RT) data, however, given the lower quality of RT data, our analysis is based only on available DM data.

Our model was trained using data from the Mediterranean basin collected between 2015 and 2020. To ensure the data's reliability, we only selected profiles that were marked with quality flags (QFs) of 1, 2, or 8 for variables such as temperature, salinity, nitrate, and chlorophyll. Furthermore, we applied a preprocessing step on the bbp700 variable based on the study by Dall'Olmo et al. (2022), which introduced a new set of real-time quality-control tests for this variable, since no procedures have been agreed upon so far to quality control bbp700 data in real-time. Specifically, we applied three of the procedures introduced in this work to bbp700 profiles: the missing-data test, which detects and flags profiles that contain a substantial amount of missing data; the high-deep-value test, which flags profiles with unusually high bbp700 values at depth; and the negative-BBP test, which flags data points or profiles with negative bbp700 values.

## 3    PPCOn: Profile Prediction Convolutional Neural Network

Within the realm of DL, CNN has emerged as a prominent architecture (Albawi et al. (2017)). A CNN network functions as a feedforward NN capable of capturing data features through convolutional structures (Li et al. (2021)). While the two-dimensional (2D) CNN architecture is designed to extract spatial features from two-dimensional data such as images (O'Shea and Nash (2015)), the 1D CNN is specifically designed to extract temporal features from one-dimensional sequential data such as signals or time series data (Tang et al. (2020)). Due to their streamlined and efficient configuration that employs only 1D convolutions (scalar multiplications and additions), 1D CNNs offer advantages in terms of real-time processing and cost-effectiveness for hardware implementation. (Kiranyaz et al. (2021)).

This section introduces the PPCon architecture, which is primarily a 1D CNN with additional MLPs employed to transform punctual data into a vectorial shape - necessary for the training of the convolutional component. The input variables for PPCon include sampling data, geolocation, temperature, salinity, and oxygen, while the output variables comprise vertical profiles for nitrate, chlorophyll, and BBP. Despite using the same architecture, a separate model is trained for each output variable, and different hyperparameters (number of epochs, weights of the loss function, and so on) are set for each of them. This separate





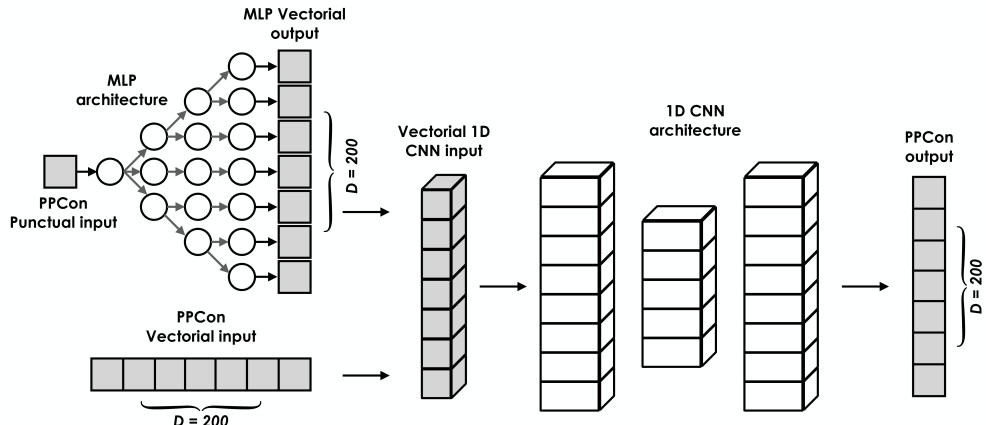

**Figure 1.** Illustration of the PPCon architecture.

tuning is necessary due to some intrinsic differences such as the numerosity of the training set and the variable ranges. The hyperparameters are tuned manually by comparing performance on the test set composed of unseen data, based on a fitness metric to be introduced later. A specific loss function is designed to promote good performances, generalization capabilities, and smooth predictions.

## 3.1 Input preprocessing

**Table 1.** The MLP component of the PPCon model illustrated in diagram form. All the 4 MLPs used in the PPCon architecture share the same architecture.

| Layer | Output Size | Activation Function |
|--------|-------------|---------------------|
| Input | $[32, 1]$ | - |
| Linear | $[32, 80]$ | SELU |
| Linear | $[32, 140]$ | SELU |
| Linear | $[32, 200]$ | SELU |
| Output | $[32, 200]$ | - |

The data considered for feeding the DL architecture comprises a collection of measurements, where each input-output pair consists of the information collected by a singular float profile. The inputs consist of two distinct categories of data, namely punctual and vectorial. Punctual data encompasses temporal and geospatial parameters, such as the sampling date (specifically year and day), geolocation, and geographic coordinates (latitude and longitude), while vectorial data encapsulates profiles of temperature, salinity, and oxygen, as recorded by the float instruments. Given that the 1D CNN architecture operates only on 135 vectorial input data, a coherent transformation of punctual features into vectorial ones is required.





In this regard, we leverage an MLP architecture that accepts punctual input and transforms it into vectorial form. MLPs are employed in order to enable the NN to automatically learn how to weigh differently the importance of such punctual input features in correspondence of different levels of depth. A separate MLP is trained for each of the four pointwise inputs. The
MLP architectures have the same number of layers and neurons contained in these layers (Table 1), since there are no a priori reasons to make them different.

During training, the weights of the MLP are optimized along with the weights of the 1D CNN architecture. Since the MLP operates as a non-linear function, this training approach enables the creation of a mapping between punctual input and its vectorial equivalent. This enables PPCon to effectively exploit punctual information and achieve optimal learning outcomes.
The output vectors generated by the MLP are concatenated with the remaining vectorial input, yielding a seven-channel tensor that serves the input of the PPCon architecture.

## 3.2 PPCon Architecture

**Table 2.** The convolutional component of the PPCon model illustrated in diagram form. The key attributes of the NN are outlined, encompassing parameters, output size (represented as [batch size, number of channels, input length]), as well as any additional layers. More specifically, "BN" denotes the batch normalization layer, "SELU" represents the non-linear selu() activation layer, and "Dropout" indicates the presence of a dropout layer along with the corresponding dropout rate. )

| Layer | Kernel | Stride | Padding | Output Size | Additional Details |
|-------|--------|--------|---------|-------------|--------------------|
| Input | - | - | - | $[32, 7, 200]$ | - |
| Conv. 1D | 2 | 1 | 2 | $[32, 64, 203]$ | BN, SELU, Dropout (rate: $d_r$) |
| Conv. 1D | 2 | 2 | 1 | $[32, 128, 102]$ | BN, SELU, Dropout (rate: $d_r$) |
| Conv. 1D | 4 | 1 | 1 | $[32, 128, 101]$ | BN, SELU, Dropout (rate: $d_r$) |
| Conv. 1D | 4 | 1 | 2 | $[32, 128, 102]$ | BN, SELU, Dropout (rate: $d_r$) |
| Deconv. 1D | 2 | 2 | 2 | $[32, 128, 200]$ | BN, SELU, Dropout (rate: $d_r$) |
| Conv. 1D | 3 | 1 | 1 | $[32, 128, 200]$ | BN, SELU, Dropout (rate: $d_r$) |
| Deconv. 1D | 2 | 2 | 1 | $[32, 64, 398]$ | BN, SELU, Dropout (rate: $d_r$) |
| Conv. 1D | 2 | 2 | 1 | $[32, 32, 200]$ | BN, SELU, Dropout (rate: $d_r$) |
| Conv. 1D | 3 | 1 | 1 | $[32, 1, 200]$ | BN, SELU |
| Output | - | - | - | $[32, 1, 200]$ | - |

The convolutional component of the PPCon architecture, summarized in Table 2, is a DL model comprising multiple 1D convolutional and deconvolutional layers.
The input tensor has a 1-dimensional shape, with a total number of channels equal to 7, one for each of the three variables to reconstruct.

The architecture includes a total of nine layers, each of which applies a set of filters to the input tensor. These filters are designed to detect specific features or patterns, with the number and size of the filter kernels specified by the parameters of





each layer. To enable effective feature extraction across different scales, various stride parameters are employed to specify the
step size at which the filters are applied to the input tensor. To ensure that the output tensor has the same shape as the input
tensor, padding parameters are incorporated, adding zero padding to the borders of the input tensor. The output tensor is then
normalized through a batch normalization (Santurkar et al. (2018)) layer after each convolutional layer. The normalization
process ensures that the output tensor has a mean of zero and a unit variance, thereby minimizing the effect of covariate
shifts and enhancing the stability of the training process. Following normalization, the output tensor is passed through a scaled
exponential linear unit (SELU) activation function (Rasamoelina et al. (2020)), which is defined as:

$$f(x) = \begin{cases} \lambda x & \text{if } x \geq 0 \\ \lambda \alpha(e^x) & \text{if } x < 0 \end{cases} \tag{1}$$

where and $\lambda \approx 1.0507$ and $\alpha \approx 1.6732$. SELU has been selected as an activation function as it induces self-normalization
properties. Dropout layers (Baldi and Sadowski (2013)) are also incorporated to prevent overfitting during training, promoting
robust generalization and enhancing the NN ability to learn diverse features from the input data. These layers randomly drop
out some of the network neurons, with the specific probability of dropout ($d_r$) specified for each layer in the architecture's
hyperparameters.

The final convolutional layer produces a 1-channel output tensor, which represents the final prediction of the model.

### 3.3   Loss function

The choice and design of a loss function is a crucial step in the development of DL models, as it determines the objective to be
optimized during training and can have a significant impact on the model's ability to generalize to new data. Besides the ability
to skilfully reproduce output variable profiles, we want the PPCon architecture to mitigate overfitting and produce a smooth
prediction curve.

To fulfil these objectives, we define a loss function comprising three components: first, the Root Mean Square Error (RMSE)
between the target output and the PPCon architecture's prediction, to assess prediction quality. Second, to mitigate overfitting
phenomena, a regularization term known as $\lambda$-regularization is employed, which penalizes complex curves in proportion to
the square of the model's weights (Zou and Hastie (2005)). By promoting smaller weight values, this technique encourages
the generation of more general predictions. The severity of this penalty is determined by a multiplicative factor $\lambda$, which is a
hyperparameter of the model. The final component of the loss function is incorporated to promote the generation of a smoother
output curve. This term, controlled by a hyperparameter $\alpha$, serves as a regularization technique that penalizes sharp variations
in the output. The final loss formula is as follows:

$$\mathcal{L}(y, \hat{y}) = \sum_{i=1}^{n}(y_i - \hat{y}_i)^2 + \lambda \sum_{i=1}^{N}|\theta_i| + \sum_{i=1}^{n-1}(\hat{y}_{i+1} - \hat{y}_i)^2 \tag{2}$$

where $y$ represents the target value, $\hat{y}$ the output of PPCon model, $n$ the length of both $y$ and $\hat{y}$ and $N$ the total number of
weights of the DL architecture.



## 4 Experimental Study

This section presents the experimental settings for the PPCon architecture, which are defined for each predicted variable under consideration. The complete code for the reproducibility of the results presented in this paper is available at: https://zenodo.org/record/8369573.

### 4.1 Training

We divided the dataset into three subsets: training, testing, and validation. The training set was used for model training and parameter optimization. The testing set was utilized to evaluate the model's performance on unseen data and assess its generalization ability. Finally, the validation set was employed for hyperparameter tuning and model selection. The dataset was randomly partitioned, ensuring that each subset contained a representative distribution of the overall data characteristics. The sizes of the training, testing, and validation sets were chosen as 80, 10, and 10. This approach enabled us to assess and validate the performance of our model effectively. Moreover, before operating this partition, a few float instruments have been selected and all of their measurements have been excluded from both the training, test, and validation set. These samples will be used as an external validation dataset. The metrics and the performances over this external validation dataset are a more effective indicator of the generalization capabilities of the PPCon model with respect to the metrics on the test set.

In order to train the NN model efficiently, the input dataset is partitioned into minibatches, where each minibatch contains 32 samples. The batch size, which determines the number of samples processed before updating the model weights, is a hyperparameter that must be set prior to training. By processing multiple samples in a minibatch, the model can update its parameters more frequently, which can lead to faster convergence and improved generalization performance (Bottou (2010)). Once all the mini-batches have been processed by the optimization algorithm, the model has completed an epoch of training.

Adadelta (Zeiler (2012)) is the algorithm that is selected as the optimizer for training the network due to its ability to dynamically adapt over time using only first-order information. This method eliminates the need for manual tuning of the learning rate and has been found to exhibit robustness in the face of noisy gradient information, various data modalities, different model architecture choices, and hyperparameter selection.

It is worth recalling that the PPCon architecture includes a 1D CNN and four MLPs, which convert point-wise input into a vector form suitable for use by the CNN. The MLPs and the CNN component of PPCon were trained using the same optimizer, with concurrent weight updates across all networks. This approach enables the joint learning of optimal information transfer from point-wise input to vector form, as well as the accurate generation of predicted profiles based on the input tensor.

To accelerate the training process, the model was trained using a GPU (graphics processing unit), which allowed for parallelized computation of the forward and backward passes.

The model's performance was evaluated once every 25 epochs by assessing its ability to predict outcomes on the test set, which consists of previously unseen data. To prevent overfitting and minimize computational burden, we introduced an early stopping routine. Specifically, the training was interrupted if the error metrics on the validation set increased for two consecutive



|  | #samples | #epochs | batchsize | dropout rate | $\lambda$ | $\alpha$ |
|---|---|---|---|---|---|---|
| nitrate | 2337 | 50 | 32 | 0.2 | 0.001 | 0.001 |
| chlorophyll | 3189 | 150 | 32 | 0.2 | 0.0001 | 0.0001 |
| BBP | 3942 | 100 | 32 | 0.2 | $1e^{-7}$ | $1e^{-7}$ |

**Table 3.** Summary of hyperparameters and dataset sizes for all inferred variables.

evaluations (i.e. after 50 epochs of training). The final model selected was the one trained prior to the two consecutive test loss increases.

## 4.2 Experimental Settings

Since each output variable has intrinsic differences in training set size, range of values, and profile shapes and variabilities, a separate hyperparameter tuning step is performed for each of them. These hyperparameters were tuned using a systematic search over a range of values, guided by the performance of the model on a held-out validation set. To avoid overfitting in the test set, we employed cross-validation techniques to estimate the generalization performance of the model and selected the hyperparameters that yielded the best performance.

The hyperparameters used for training the three PPCon architectures are summarized in Table 3, together with the size of the dataset, the total number of epochs performed, and the batch size dimension which have already been discussed in previous sections.

In our experiments, we applied a dropout rate of 0.2, which was consistent across all trained models. This means that during training, each neuron in the NN has a 20% chance of being randomly excluded from the computation. Dropout regularization is a technique used to prevent overfitting by encouraging each neuron to encode information independently, thereby inhibiting co-dependencies among neurons.

Table 3 also reports the multiplicative factors that determine the relative contributions of different elements that compose the loss function defined in Section 3.3. The values of these hyperparameters vary depending on the variable being inferred, as these variables have different orders of magnitude and result in RMSE values that vary in magnitude as well. It is crucial to accurately balance the regularization term, governed by $\lambda$, and the smoothness term, governed by $\alpha$, to prevent them from dominating the loss function's RMSE component. The optimal values reported in Table 3 guarantee a good and smooth prediction of the vertical profile.

The last implementation detail to be addressed concerns the creation of vectors used to feed the PPCon architecture. As previously discussed, vectorial inputs of different natures are fed into the CNN component of PPCon: firstly, the outputs of an MLP architecture; secondly, vectors representing input variables (temperature, salinity, oxygen) at different depths. To ensure that all input vectors have the same length, we adopted the following strategy: (i) the output and input variables have been interpolated on a regular grid of size 200, and (ii) the output of MLPs have the same length and discretization of the input variable vectors. For nitrate, we considered a depth range of 0 to 1000 meters with an interpolation interval of 5 meters,



|  | North West Med. | South West Med. | Tyrrhenian | Ionian | Levantine |
|---|---|---|---|---|---|
| Latitude | $40°N - 45°N$ | $32°N - 40°N$ | $37°N - 45°N$ | $30°N - 45°N$ | $30°N - 37°N$ |
| Longitude | $-2°E - 9.5°E$ | $-2°E - 9.5°E$ | $9.5°E - 15°E$ | $14°E - 22°E$ | $22°E - 36°E$ |

**Table 4.** Geographical limits of the five areas in which the Mediterranean is divided for the posterior analysis.

| nitrate | | | | chlorophyll | | | | bbp700 | | | |
|---|---|---|---|---|---|---|---|---|---|---|---|
| WMO | date | lat | lon | WMO | date | lat | lon | WMO | date | lat | lon |
| 6901653 | 24/07/2014 | 41.85 | 4.55 | 6902901 | 10/10/2019 | 42.91 | 7.62 | 6901657 | 13/02/2019 | 39.80 | 7.23 |
| 6901769 | 20/12/2016 | 39.11 | 10.96 | 6901776 | 13/04/2014 | 42.79 | 7.01 | 6901776 | 20/09/2014 | 42.76 | 7.25 |
| 6901771 | 18/10/2015 | 36.01 | 20.12 | 6901773 | 19/09/2017 | 32.93 | 31.24 | 6903240 | 20/05/2018 | 42.20 | 28.57 |

**Table 5.** WMO, date, and geolocation of the float profiles reported in Figure $1 - 3$.

whereas, for chlorophyll and BBP, we considered a depth range of 0 to 200 meters with an interpolation interval of 1 meter. Then, we set the output layer dimension of the MLP to 200 to ensure that all input vectors have the same length. As a result, the

245    final dimension of the input tensor is 7 (the number of inputs) x 200 (the length of the input vector) x the number of dimensions in the training set.

## 4.3   A posterior validation analysis of PPCon

To validate the PPCon architecture, we conducted a thorough analysis of its average performance in different geographic areas and across various seasons. The choice underlying this investigation originates from the fact that diverse geographic areas and

250    seasons are known to have distinct profile properties, such as the shape of the vertical profiles (e.g. depth and slope of the nitracline, or depth and intensity of DCM) and the values at the surface and in deep water. Our goal is to evaluate the model's ability to accurately capture these variations. We wish to once again note the model is trained on the entire dataset, and this division is purely for a posteriori analysis of the performances. This a posteriori analysis of the performances has the objective of identifying possible influence and bias of the uneven spatial and temporal distribution of the profiles on the performance of

255    the PPcon model.

     Five different geographic areas are considered, namely: Northern West Mediterranean, Southern West Mediterranean, Tyrrhenian, Ionian, and Levantine. The geographical limits related to these variables are reported in Table4.

     To gain a comprehensive understanding of how the model performs across different seasons, we analyzed four distinct time periods: winter (January to March), spring (April to June), summer (July to September), and autumn (October to December).




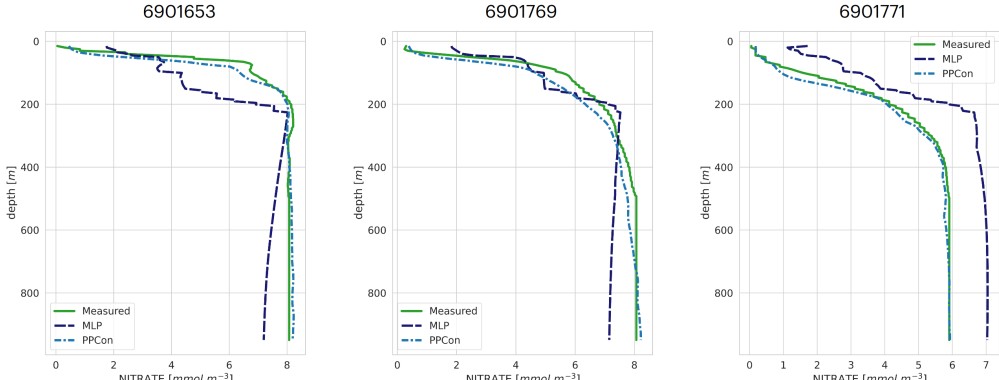

**Figure 2.** Profiles of nitrate for some selected floats (WMO numbers in the title) and dates. Comparison between measured profile (green lines), MLP reconstruction (azure dashed lines), and PPCon reconstruction (blue dash-dotted lines). Profiles are from the subset used for the test.

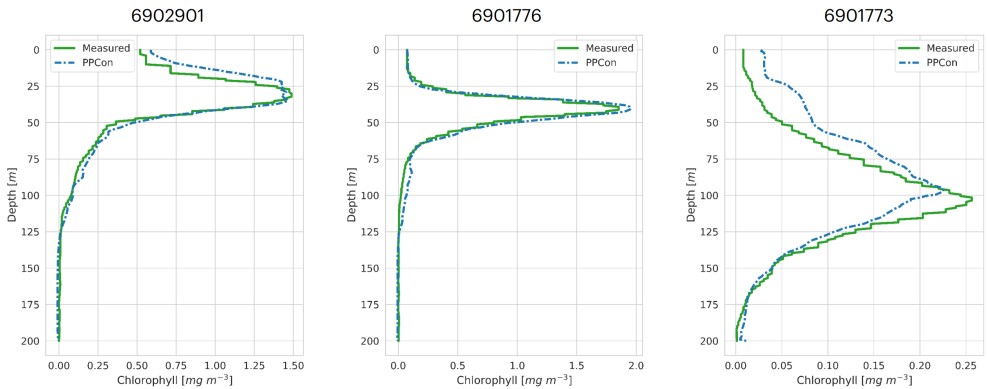

**Figure 3.** Profiles of chlorophyll for some selected floats (WMO numbers in the title) and dates. Comparison between measured profile (green lines) and PPCon reconstruction (blue dashed lines). Profiles are from the subset used for the test.

| | Winter | | Spring | | Summer | | Autumn | |
| --- | --- | --- | --- | --- | --- | --- | --- | --- |
| | train | test | train | test | train | test | train | test |
| Nitrate | 0.640 | 0.665 | 0.591 | 0.610 | 0.570 | 0.615 | 0.631 | 0.640 |
| Chlorophyll | 0.082 | 0.098 | 0.129 | 0.134 | 0.069 | 0.067 | 0.045 | 0.045 |
| bbp700 | $2.561e^{-4}$ | $2.571e^{-4}$ | $3.101e^{-4}$ | $2.691e^{-4}$ | $1.871e^{-4}$ | $1.941e^{-4}$ | $2.211e^{-4}$ | $1.971e^{-4}$ |

**Table 6.** RMSE calculated between the float measurements and the reconstructed values obtained from the PPCon architecture for all variables inferred. This metric is evaluated individually for the train and test sets. The RMSE is computed for different seasons of the year (described in Section 2).





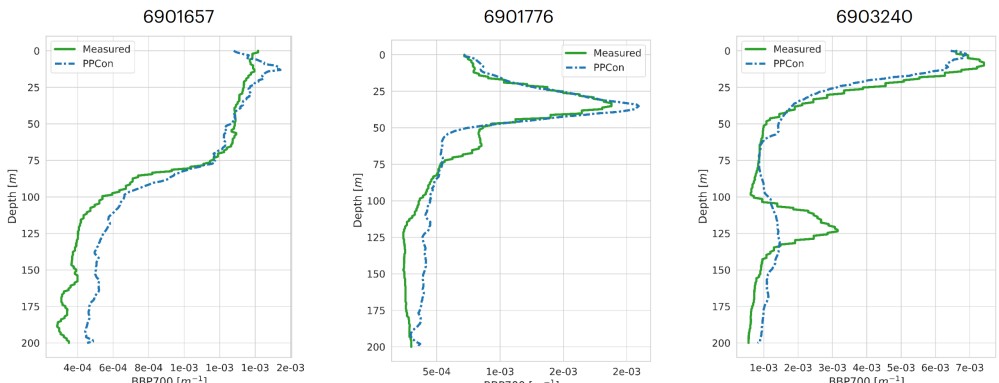

**Figure 4.** Profiles of bbp700 for some selected floats (WMO numbers in the title) and dates. Comparison between measured profile (green lines) and PPCon reconstruction (blue dashed lines). Profiles are from the subset used for the test.

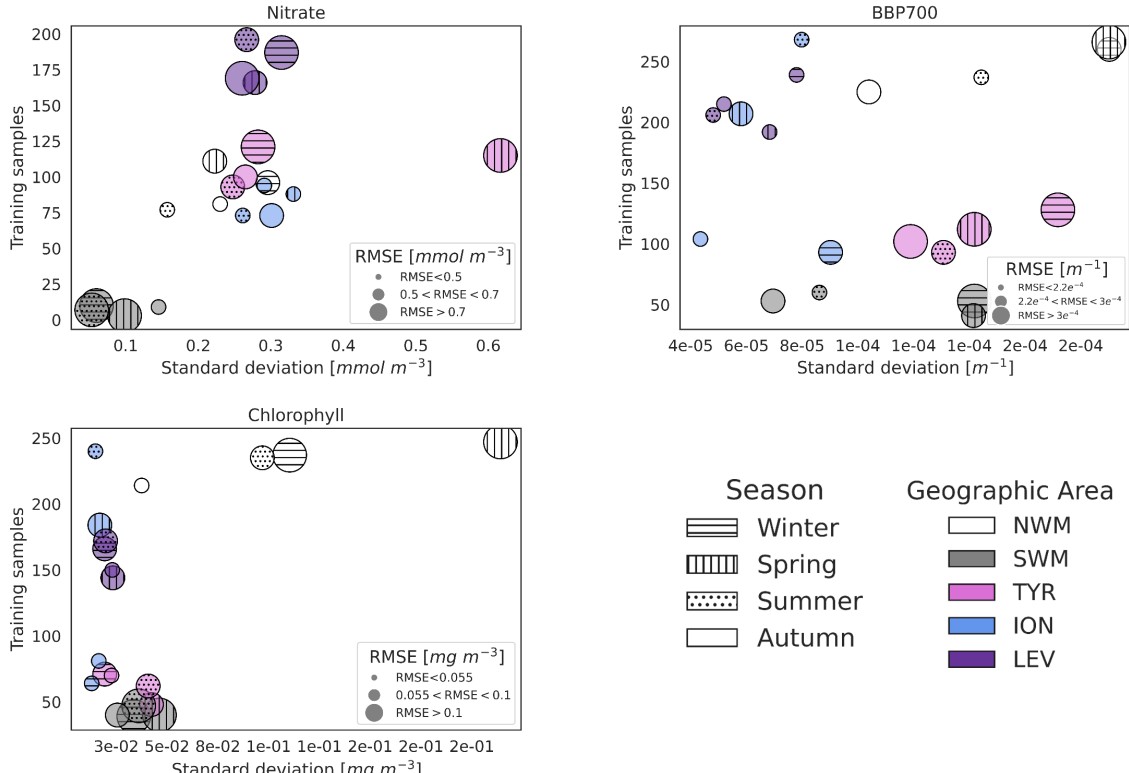

**Figure 5.** Plot of the RMSE distribution with respect to the data variability and training dataset size of the different sub-areas (colour of the symbol) and the different seasons (symbol fill pattern). RMSE values are categorized by the size of the symbols.





| | North Western Med | | South Western Med | | Tyrrhenian | | Ionian | | Levantine | |
|---|---|---|---|---|---|---|---|---|---|---|
| | train | test | train | test | train | test | train | test | train | test |
| Nitrate | 0.514 | 0.531 | 0.637 | 0.643 | 0.706 | 0.704 | 0.426 | 0.445 | 0.680 | 0.718 |
| Chlorophyll | 0.119 | 0.139 | 0.103 | 0.098 | 0.074 | 0.075 | 0.051 | 0.047 | 0.055 | 0.051 |
| bbp700 | $2.661e^{-4}$ | $2.641e^{-4}$ | $2.501e^{-4}$ | $2.581e^{-4}$ | $3.871e^{-4}$ | $2.831e^{-4}$ | $2.051e^{-4}$ | $2.081e^{-4}$ | $1.731e^{-4}$ | $1.791e^{-4}$ |

**Table 7.** RMSE calculated between the float measurements and the reconstructed values obtained from the PPCon architecture for all variables inferred. This metric is evaluated individually for the train and test sets. The RMSE is computed for different geographic areas (described in Section 2).

## 5 Results

This section presents the results of the PPCon model in predicting nitrate, chlorophyll, and bbp700 profiles. The effectiveness of the model is evaluated by presenting both quantitative skill metrics (i.e., RMSE) and visual representations of the predicted profiles based on the test set.

Specifically, we assess the PPCon performance over different seasonal variations (Table 6), and different geographic areas (Table 7). The absence of overfitting is supported by reporting the RMSE for both the training and test sets, which exhibit non-dissimilar values.

In terms of performances across different geographic areas (Table 7), it can be seen that the lowest RMSE values for chlorophyll and bbp700 are in the eastern sub-basins, while for nitrate the lowest and highest values are in the two eastern sub-basins. Notably, the prediction accuracy for nitrate is significantly higher in the Ionian Basin, with RMSE values below $0.5$. Considering the temporal evolution of RMSE values (Table 6), the highest values of chlorophyll and bbp700 are in spring and winter, which appears reasonable given the higher variability of vertical pattern during these seasons (Cossarini et al. (2019); Teruzzi et al. (2021)). Errors for nitrate are fairly homogeneous among the seasons, which are the highest during winter (e.g. vertical mixing season) and the lowest values during the stratification seasons (i.e. spring and summer). As for chlorophyll, the western basin of the Mediterranean shows higher RMSE values. This can be attributed to the naturally elevated chlorophyll levels observed in that specific area, which consequently lead to higher RMSE values.

For each variable investigated, we present three instances of vertical profile reconstruction using the PPCon architecture, compared to the profile measured by the float instrument whose corresponding identification number is indicated above each profile. To ensure geographic and seasonal diversity, we have selected profiles representing different regions, including at least one from the West Mediterranean and one from the East Mediterranean. Figure 2-4 displays examples of, respectively, reconstructed nitrate, chlorophyll, and bbp700 profiles. For the nitrate variable, the reconstruction performed by the MLP model is also reported (Pietropolli et al. (2023a)). The information related to these profiles, such as the date and geolocation of sampling, are reported in Table 5.

The obtained results confirm the quality of the profiles generated by the PPCon architecture, which appears to better reconstruct the shape and smoothness of the profiles than the previous MLP architecture. Indeed, PPCon is able to capture different





| Nitrate | | Chlorophyll | | bbp700 | |
|---|---|---|---|---|---|
| 6901648 | 0,4288 | 6901648 | 0,1739 | 6901649 | $2,6231e^{-4}$ |
| 6901764 | 0,4583 | 6901496 | 0,0878 | 6900807 | $1,5071e^{-3}$ |

**Table 8.** RMSE calculated between the float measurements and the reconstructed values obtained from the PPCon architecture over the external validation dataset.

profile shapes associated with different geographic and seasonal conditions, as clearly demonstrated by the predicted nitrate and chlorophyll profiles. Higher quality in the prediction is achieved for the nitrate variable, followed by chlorophyll, and last by bbp700. This outcome is expected, as the nitrate variable exhibits lower variability in the values and profile shapes than the other two variables. For a more detailed analysis of the behavior of the PPCon architecture quality of the predicted profiles, Appendix A reports a comparison between the mean of PPCon predicted profiles and the mean of profiles measured by the

float instruments (in the test set) for all investigated variables, providing a more specific insight on the PPCon performances in different geographic areas and seasons.

      In order to understand the impact of the training set numerosity and of the variability of profiles on the quality of the PPCon predictions we investigated the relation between these quantities and the PPCon error. Specifically, Figure 5 reports points whose size corresponds to the prediction RMSE (divided according to the four seasons and the five geographic areas) and

their relation with the variability of the training set (on the $x$-axis), quantified by the standard deviation, and the numerosity of the training set (on the $y$-axis). This figure offers also valuable insight into the geographical and seasonal distribution of the training dataset dimension. The analysis of the nitrate plot reveals fairly homogeneous errors across all sub-areas and seasons, suggesting a lack of a strong relationship between errors and variability or data availability. In terms of chlorophyll and bbp700, both variables exhibit similar behavior. In particular, data availability appears to have no significant impact on the

error, whereas RMSE tends to increase proportionally with the variability.

**5.1    PPCOn performance over an external validation dataset**

For each inferred variable, Figures 6-8 display Hovmöller diagrams of measured and reconstructed float instruments belonging to the external validation set, and Table 8 reports the corresponding RMSE values. This represents a particularly stringent validation test since none of the profiles measured by these floats have been encountered by the PPCon model during the

training or validation phases. The figures compare the in situ float measurements (upper diagram) and the predictions generated by the PPCon architecture (lower diagram) for floats that have been specifically selected to cover different geographical regions of the Mediterranean Sea (e.g. one in the eastern and one in the western Mediterranean Sea). White lines in the diagram indicate float measurements that cannot be compared due to various reasons such as the sensor's temporary inability to measure the specific variable inferred or the absence of one of the inputs necessary for the PPCon architecture (e.g. at least one between

temperature, salinity, and oxygen). This could be attributed to limitations in the sensor or unacceptable quality flags associated with the collected data. Nevertheless, the number of profiles that cannot be calculated by PPCon is rather low and does not

**Figure 6.** Hovmöller diagrams for the nitrate of two selected floats (WMO name in the title) belonging to the external validation set. BGC-Argo measurements (upper panels) and PPCon prediction (lower panels) are compared. WMO 6901648 sampled the $40°N - 42°N$ and $2°E - 6°E$ area during $2014 - 2016$, whereas WMO 691764 sampled the $31°N - 34°N$ and $26°E - 40°E$ area during $2015 - 2017$.

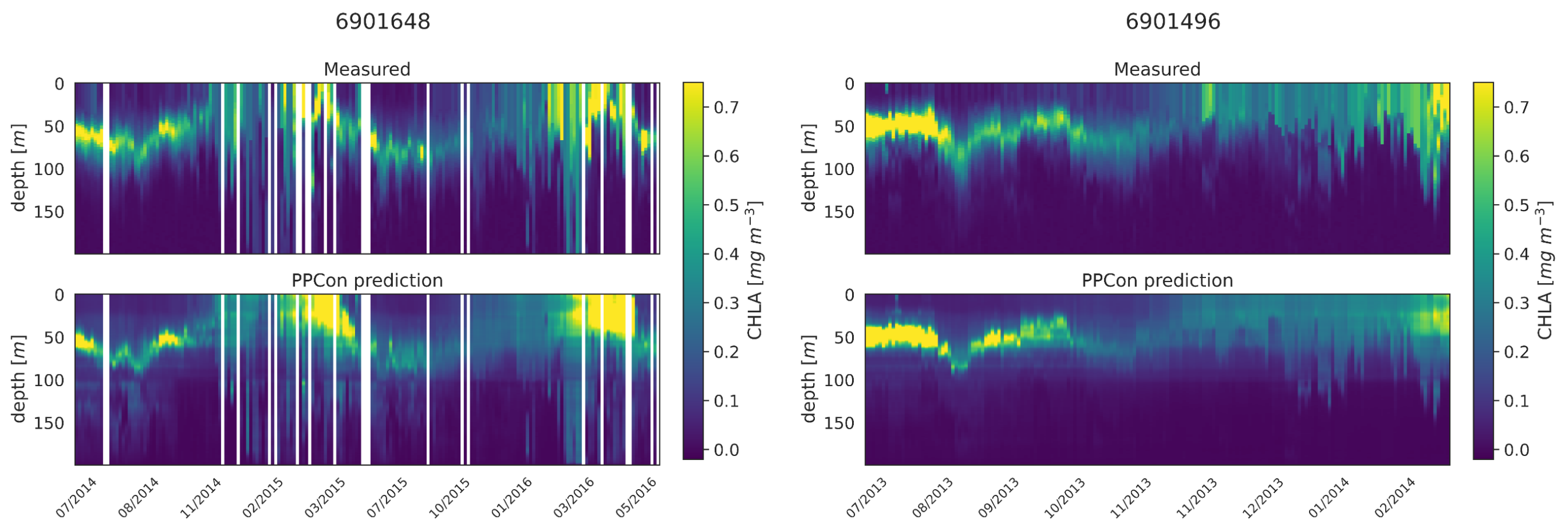

**Figure 7.** Hovmöller diagrams for the chlorophyll of two selected floats (WMO name in the title) belonging to the external validation set. BGC-Argo measurements (upper panels) and PPCon prediction (lower panels) are compared. WMO 6901648 sampled the $40°N - 42°N$ and $2°E - 6°E$ area during $2014 - 2016$, whereas WMO 6901496 sampled the $42°N - 43°N$ and $7°E - 12°E$ area during $2013 - 2014$.





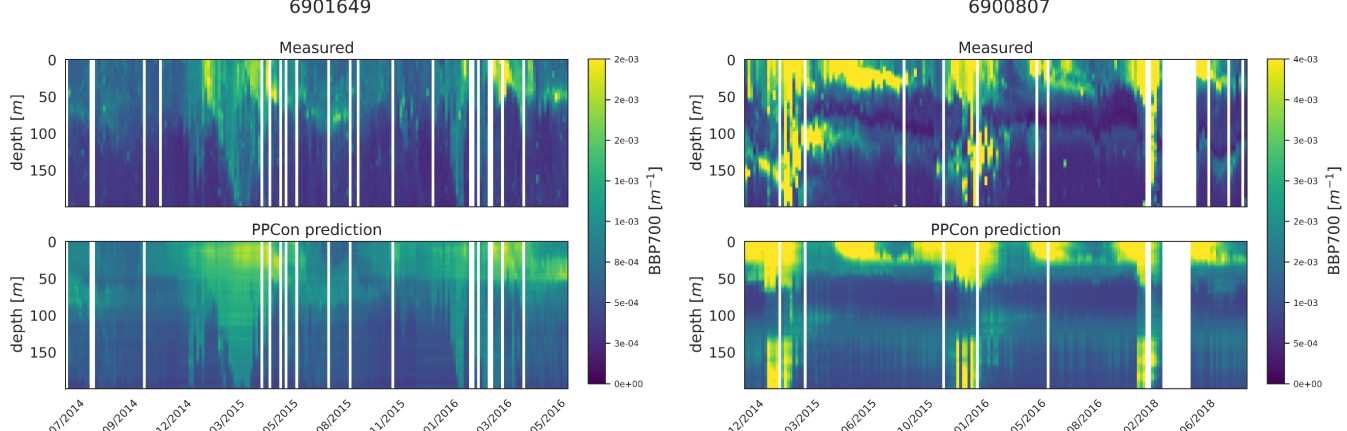

**Figure 8.** Hovmöller diagrams for the bbp700 of two selected floats (WMO name in the title) belonging to the external validation set. BGC-Argo measurements (upper panels) and PPCon prediction (lower panels) are compared. WMO 6901649 sampled the $39°N − 41°N$ and $3°E − 7°E$ area during $2014 − 2016$, whereas WMO 6900807 sampled the $41°N − 44°N$ and $29°E − 35°E$ area during $2014 − 2018$.

degrade the very good capacity of the reconstructed profiles to reproduce the temporal evolution of the vertical dynamics shown by the measured floats.

These plots confirm the PPCon capability to perform accurate predictions also regarding float devices which are totally
unseen by the model. The nitrate (Figure 6) reconstructions exhibit a very good performance of PPCon in predicting the vertical dynamics associated with the temporal evolution of the nutricline depth (i.e. the depth at which the sharp increase of the nitrate values is observed), the values in the deep layers (which are different in the sub-areas sampled by the two floats), and the occurrence of deep vertical mixing when surface concentration increases to values higher than $3mmol/m^3$. Particularly impressive is the capability of PPCon to reconstruct the temporal dynamics of chlorophyll (Figure 7). The reconstruction
effectively captures the evolution of the chlorophyll surface peaks during winter and the formation of the deep chlorophyll maximum during summer in both floats representing the two areas of the Mediterranean Sea. Among the three variables, the bbp700 (Figure 8) shows the least accurate predictions. However, the model still displays the ability to infer the key characteristics of the variable's temporal behavior. Nonetheless, the generated predictions for bbp700 appear slightly less detailed compared to the original sampling, indicating a partial limitation of the model in capturing small-scale variations.
Quantitatively, the prediction quality of the PPCon architecture (RMSE values in Table 8 are fairly well aligned with the metrics calculated over the test set, as indicated in Table 7. In particular, nitrate errors of the two floats are quite homogeneous and $30\%$ lower than the RMSE values of the test set. The errors in chlorophyll and bbp700 predictions exhibit greater variability, with values almost double for the floats in the western Mediterranean with respect to the eastern ones. This is, however, in line with results reported in Table 7, and Figure 5 where higher errors are associated with higher variability.





## 6 Discussion

To our knowledge, the PPCon architecture is the first attempt to predict vertical BGC-Argo profiles by means of a convolutional architecture. Its primary objective is the incorporation of typical profile shapes during the training phase, in contrast with previous architectures which all relied on MLP architectures and point-wise strategy. There are notable distinctions between the two approaches: MLPs were trained on cruise data, which are known to be more precise in collecting variables than autonomous sensors such as the BGC-Argo (Johnson et al. (2013); Johnson and Claustre (2016)). However, while MLP architectures can provide good training and test errors (Pietropolli et al. (2023a); Fourrier et al. (2020); Bittig et al. (2018); Sauzède et al. (2017)), they have been found to exhibit higher errors when predicting BGC-Argo profiles (Pietropolli et al. (2023a)). In contrast, the PPCon architecture, which relies directly on BGC-Argo float measurements for the training, showed very good test and external validation performances.

However, it should be noted that an intrinsic measurement error is introduced by the higher uncertainty of the variables measured throughout the autonomous sensors. We alleviated this limitation by using only DT and high-quality checked Argo and BGC-Argo floats data; however, the use of the present PPCon in operational oceanography (Le Traon et al. (2021); Cossarini et al. (2019)) should be considered cautiously given the lower reliability of Adjusted or near real-time (NRT) Argo data. According to the analysis conducted by Mignot et al. (2019), the BGC-Argo float data for oxygen, nitrate, and chlorophyll concentrations exhibit RMSE values evaluated at $5.1 \pm 0.8 \mu mol/kg$, $0.25 \pm 0.07 \mu mol/kg$, and $0.03 \pm 0.01 mg/m^3$, respectively. On the other hand, we have demonstrated that the RMSE for the PPCon architecture is $0.61$. Therefore, a research question pertains to understanding how the measurement error of the float instrument impacts the performance of the PPCon architecture, and how to estimate an overall error that combines the contribution of the instrument error and the error associated with the PPCon.

Although both MLPs and PPCon employ similar input information (date, geolocation, temperature, oxygen, and salinity), their treatment of this data differs significantly. MLPs process the input and output as discrete data points, while PPCon utilizes vector representations of the vertical profiles. This approach was necessitated to effectively exploit the potential of a 1D CNN, which intrinsically preserves the characteristic profile shape of the input and output variables Kiranyaz et al. (2021). When comparing the predictive performance of these techniques in generating vertical profiles from float data, distinct differences emerge. MLPs tend to produce profiles characterized by apparently artificial discontinuity and jumps, while the profiles generated by PPCon exhibit a smoother and more realistic appearance (Figure 2). This improvement is confirmed also by the RMSE, which is lower when using the PPCon model ($RMSE_{PPCon} = 0.61$) compared to the state of the art of MLP architectures ($RMSE_{MLP} = 0.87$ according to Pietropolli et al. (2023a)).

Moreover, the posterior study that we conducted shows that there is no significant variation in the error across different geographic areas and seasons of the year (Table 6-7), confirming that PPCon can be successfully applied to all the float profiles collected in the Mediterranean basin.

Specifically, the PPCon architecture serves as a valuable tool for significantly enriching the BGC-Argo dataset. This becomes useful as ocean observing systems - while essential for the monitoring of the health of the marine ecosystem (Euzen




et al. (2017)) - have considerable limitations given their sparse and scarce space-temporal coverage. Surface satellite observa-
tions are limited by cloud covers and incomplete swaps of satellite sensors (Donlon et al. (2012)), while profiling the ocean
interior is limited by the capacity of deploying and retrieving sensors and measurements with sufficient coverage. Gap filling
and interpolation of satellite observations (Volpe et al. (2018), Sammartino et al. (2020); Alvera-Azcárate et al. (2005)) are
nowadays consolidated practices to provide gap-free and high-level products (Barth et al. (2020); Sauzède et al. (2016)). Our
PPCon architecture presents a valuable approach to harness the potential of the Argo and BGC-Argo network by enabling the
synthetic generation of essential variables (chlorophyll, nitrate, and bbp700) even when these costly sensors are not present in
the deployed floats. The application of PPCon on the GDACs BCG-Argo float dataset (spanning from 2015 to 2020) enabled
the generation of 5234 synthetic nitrate profiles, 3879 chlorophyll profiles, and 3307 bbp700 profiles, which means doubling
the chlorophyll and bbp700 BGC-Argo profiles and more than tripling those of nitrate. Enhancing the float dataset through the
inclusion of reconstructed nutrient profiles (and possibly other biogeochemical variables) has been proved successful in observ-
ing system simulation experiments (Ford (2021); Yu et al. (2018)) and in real assimilation numerical experiments (Amadio et
al., (2023)). In particular, the assimilation of reconstructed profiles effectively corrects a widespread positive bias observed in
the Operational System for Short-Term Forecasting of the Biogeochemistry of the Mediterranean (MedBFM), and the addition
of the reconstructed profiles increases the spatial impact of the BCG-argo network from 20% to 45% (Amadio et al., (2023))

## 7 Conclusions

This paper presents a novel approach for reconstructing low-sampled variables, namely nitrate, chlorophyll, and bbp700, using
high-sampled variables such as date, geolocation, temperature, salinity and oxygen. The introduced model, named PPCon,
utilizes a spatial-aware 1D CNN architecture that effectively learns the characteristic shape of the vertical profile, enabling
precise and smooth reconstructions. PPCon represents a notable advancement over previous techniques relying on MLPs, which
operate on point-to-point input and output, making it challenging to generate continuous curves when forecasting complete
vertical profiles. The training dataset consists of a collection of BGC-Argo float measurements in the Mediterranean basin. The
proposed architecture has been specifically designed to handle both punctual and vectorial input, with careful tuning of the
architecture and loss function for the task. An extensive hyperparameter tuning phase has been conducted to ensure the best
architecture for each variable.

To evaluate the accuracy of the profiles generated by the PPCon architecture, both quantitative metrics and visual repre-
sentations of the results have been provided. Additionally, the method has been validated on an external dataset to verify its
generability. The results confirm the model's ability to predict high-quality synthetic profiles, with particularly accurate predic-
tions for the nitrate variable, followed by chlorophyll, and lastly, bbp700. The RMSE for nitrate reconstruction is reduced from
$RMSE_{MLP} = 0.87$ to $RMSE_{PPCon} = 0.61$. PPCon demonstrates its capacity to capture and learn distinct typical shapes
in the profiles, which characterize the inferred variables across different seasons and geographic areas. Detailed error analysis
confirms the model's robust performance, accounting for seasonal and regional variations, suggesting that PPCon's ability to





learn these differences can make it successful for broader-scale training beyond the Mediterranean basin. Furthermore, the model exhibits accurate performance on an external validation dataset, confirming its potential for generalization.

*Code and data availability.* The datasets, source code, and model implementation used in this study are publicly available at https://github.com/gpietrop/PPCon for interested readers to access and replicate the results presented in this paper (Pietropolli et al. (2023b)).

400 **Appendix A: Extended Results**

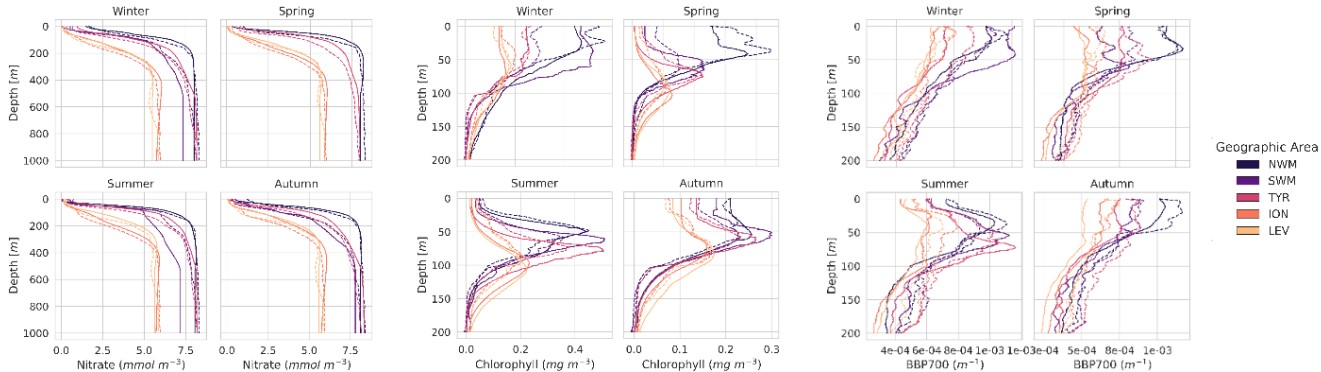

**Figure A1.** Comparison of the mean of PPCon predicted profiles with the mean of sampled values measured by the float instruments in the test set, for all the variables investigated. The mean is computed over all the profiles belonging to a fixed geographic area and a fixed season.

Figure A1 presents a comparison between the mean values of the PPCon predicted profiles and the mean values of the sampled measurements obtained from the float instruments in the test set. This comparison encompasses all the variables examined. The mean values are computed based on the profiles within a specific geographic area and season. These profiles serve as additional indicators to assess the reliability of predictions within different frameworks, providing valuable insights 405 into the precision of predictions at various depth levels. These results confirm the previous observations discussed in Section 5, particularly the finding that the prediction quality is superior for the nitrate variables, followed by chlorophyll, and lastly, bbp700. Additionally, an interesting characteristic of the PPCon prediction is its higher quality in deep water compared to surface water. This can be attributed to the higher variability of profiles in the surface water, making it more challenging for the neural network to accurately capture the diverse shapes.

410 *Author contributions.* The authors contributed to this work as follows: GP, LM, and GC conceptualized the research, GP conducted data curation and analysis, GP developed the computational model, GP and GC contributed to the experimental design and methodology, LM and



GC provided critical insights and supervision, GP performed validation experiments. All authors reviewed and approved the final version of the paper.

*Competing interests.* The authors declare they have no conflict of interest.

*Acknowledgements.* Data were collected and made freely available by the International Argo Program and the national programs that contribute to it (https://argo.ucsd.edu, https://www.ocean-ops.org). The Argo Program is part of the Global Ocean Observing System (Argo (2000)).

This research has been partly supported by the MED-MFC "Mediterranean Monitoring and Forecasting Centre" of CMEMS, which is implemented by Mercator Ocean International within the framework of a delegation agreement with the European Union (Ref. n. 21002L5-COP-MFC MED-5500).



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
