# Peer review of "PPCon 1.0: Biogeochemical Argo Profile Prediction with 1D Convolutional Networks"

_EGUsphere, 2023_

## Author Comment (AC1)

**PPCon 1.0: Biogeochemical Argo Profile Prediction with 1D Convolutional Networks**

**Response to the Editor**

Dear Editor and Reviewers,
We appreciate the constructive comments and suggestions provided by the two reviewers. Following their suggestions, we propose several major changes to our manuscript, which are outlined below:

1) improving the traceability and presentation of the dataset used in our analysis:

- We have simplified and improved the description of the BGC-Argo, including its source, and added the corrected BGC-Argo DOI for better reference.
- The specific dataset employed in our analysis has been published on Zenodo (doi.org/10.5281/zenodo.10391759) to ensure traceability and repeatability of our results.
- We have more clearly specified our focus on the Mediterranean Sea in abstract and introduction.
- A new figure depicting the spatial distribution of profiles in the Mediterranean Sea has been added (Figure "New figure 2").

2) Improving readability of the text in Sections 4, 5, and 6.

3) improving the discussion on the benefits of the convolutional approach over point-wise trained neural network architectures with the addition of the new Appendix B, which compares profile reconstruction using different neural network architectures and the revision of some sentences in the discussion section.
Below, we present our point-by-point responses to the reviewers' comments. The reviewers' comments are highlighted in blue, followed by our responses in black. In each response, we detail the proposed changes to the manuscript, including any modified text and/or figures (in red).

Furthermore, due to the improvements in the selection and description of the dataset (point 1), we decided to re-run the training/test/validation. The results remain largely unchanged; minimal variations are attributable to the random selection of profiles in the training (80%), test (10%), and validation (10%) quotas. Figures have been redrawn and are presented at the end of this letter. The conclusions of our paper remain consistent with the original version.

**Reviewer 1**

The paper demonstrates the application of a convolution neural network (coupled to a MLP) to predict nitrate, chlorophyll and backscattering (bbp700) from temperature, salinity, and oxygen as well as time and geographic coordinates. This work is an improved version of a previous work by the same authors published in 2023 in Applied Sciences (https://doi.org/10.3390/app13095634 )

We appreciate the constructive comments and suggestions from the Reviewer. We present our point-by-point responses to the Reviewer's comments below. The Reviewer's comments are in blue, our responses follow each comment in black. In each response, we detail the changes we propose to make to the manuscript in order to address the following comments, and include the proposed modified text and/or figure (in red).

A. My main comment regarding this paper, one would either go into more detailed comparison between the previous published method and for example show some example profiles where they provide a different answer and discuss them. So far the comparison is mostly limited to the RMSE error. Or the method should be compared to another reconstruction technique altogether. In all cases, one should use the same training/test/validation dataset (I assume that this is already the case, please confirm).

We appreciate the comment and we will include a better comparison between PPCon and previous published methods. In doing that, it is important to note that while PPCon has been trained/tested on profile data, other previously published approaches have been trained/tested on point-wise data and subsequently used to reconstruct profiles.
Thus, we have decided to use the subset of the test dataset (10% of BGC-Argo profiles) to assess the PPCon architecture and the previous NN methods. A new appendix with reconstructed profile examples and skill performance metrics will be dedicated to the comparison and the main text will be modified accordingly.
The new appendix will be as follows:

**"Appendix B: Comparison between reconstructed nitrate profiles by PPCon and MLP architectures**

The present appendix aims to show the performance of three different ML architectures to reconstruct nitrate profiles that use Argo profiles of temperature, salinity and BGC-Argo profiles of oxygen. The three ML architectures are: the 1D CNN of the present work (PPCon), MLP trained on point-wise data from Emodnet (MLP, Pietropolli et al., 2023) and MLP trained on point-wise data (CANYON-Med, Fourier et al., 2020). The comparison is done on the sub-set of profiles used in the test phase described in section 2. Figure B1 shows some measured and reconstructed float profiles. The visual comparison reveals the higher performance of PPCon to match the shape of the measured profiles (e.g., depth and intensity of the nitracline) and the nitrate values of the deepest part of the profiles observed in the different Mediterranean sub-regions

The quantitative assessment of the performance of the three ML architectures is shown in Table B1 that reports the RMSE computed over all profiles of the sub-set used in the test

phase. RMSE of the reconstructed profile by PPCon is more than 30% lower than that computed on the MLP reconstructions.

[Figure]

Fig. B1: Nitrate profiles from BGC-Argo dataset (green, measured) and reconstructed by PPCon (cyan dashed line), MLP as in Pietropolli et al. (2023) (purple dashed line) and CANYON-Med (dark blue dashed line). Profiles are selected from the sub-set used in the test phase of the present work. Float positions are as follows: 6901032 in NWM, 6903249 and 6903153 in ION, 69032904 in LEV and 6901767 in TYR and 6901769 in SWM.

| | PPCon | CANYON-Med (Fourier et al., 2020) | MLP (Pietropolli et al., 2023) |
|---|---|---|---|
| Nitrate RMSE [mmol/m3] | 0.52 | 0.78 | 0.98 |

Table B1: RMSE of the three ML architectures computed over the nitrate profiles of the sub-set BGC-Argo dataset of the test phase."

Additionally, as reported above, old line 350-358 will be changed as follows:

"Although both MLPs and PPCon employ similar input information (date, geolocation, temperature, oxygen, and salinity), their treatment of this data differs significantly. **While the current MLP applications** process the input and output **as point-wise data** discrete data

, PPCon utilizes vector representations of the vertical profiles. This approach  effectively exploit**s** the potential of a 1D CNN, which intrinsically preserves the characteristic profile shape of the input and output variables Kiranyaz et al. (2021). When comparing the predictive performance of these techniques in generating vertical profiles from float data, distinct differences emerge. MLPs **can**  produce profiles **affected by**  artificial discontinuity , while the profiles generated by PPCon exhibit a smoother and more realistic appearance (**Appendix B2**). **Additionally**,  **the RMSE values computed on the reconstructed nitrate profiles of the test sub-set confirms the better performance of the 1D CNN approach with respect to a MLP approach trained on point-wise data** (**Appendix B2**). "

Finally, old line 384 in conclusion will be changed as follow:

"The introduced model, named PPCon, utilizes a spatial-aware 1D CNN architecture that effectively learns the characteristic shape of the vertical profile, enabling precise and smooth reconstructions. PPCon represents a **potential**  advancement **in predicting BGC-Argo profiles** over previous  **on MLP applications,** which operate on point-wise  input and output"

B. In general the structure of the neural network is rather "unorthodox". It would be good to justify the approach (in particular the secession of convolutional layers; see below).

The structure of the network was chosen to process both 1D inputs and scalar values. Hence the choice of a CNN to process the multiple channels of 1D inputs and produce the corresponding 1D outputs. To integrate scalar values a MLP composed only of fully connected layers has been employed to produce, for each scalar value, an additional input channel. A more in detail justification for the kernel sizes choice and the use of deconvolutional layer is provided below in response to a more specific comment of the reviewer addressing this concern.

C. MLP vs CNN: the description is a bit shallow as mathematically a CNN is a special case of a MLP in the sense that a convolution operation is a special case of a matrix multiplication. CNNs assume translation invariance as an additional prior information, which is why they can outperform MLPs (when considering multiple channels a 1x1 convolution is actually an MLP over the channel dimension).

It is correct that CNNs are a special case of MLP where additional constraints are imposed (namely translational invariance). It is also correct that the additional prior information is the reason CNN are used instead of working only with fully connected layers in MLP, since it is not necessary to learn the already known property of translational invariance which is instead encoded in the architecture. That is also the reason for our choice of 1D CNNs, since (some of) the inputs have an inherent spatial dimension (i.e., the vertical profiles).

It is correct that the term MLP (contrasted with CNN) was used without the necessary adherence to the formal definition, since it is also of common usage to identify MLP with

"sequence of fully connected layers". We will clarify the distinction between CNN and MLP and we will be careful to avoid the more informal use of MLP in the text.

It is correct that a CNN can be mathematically considered a specialized form of an MLP, where the convolution operation represents a specific case of matrix multiplication, as discussed in Goodfellow et al. (2016). This perspective highlights the fundamental similarities between the two types of neural networks.

However, the distinction lies in the additional intrinsic property of translation invariance inherent in CNNs, which also holds for 1D CNNs, as used in our study.
This property is crucial when dealing with vector inputs, such as signals, where there is a correlation between points in the vector. And this is exactly our case study, in fact where the vector represents the value measured in the vertical profile.
CNNs are adept at capturing local dependencies and identifying invariant features, making them highly effective for tasks involving sequential or spatially-correlated data.

In summary, while acknowledging the similarity between MLPs and CNNs, our choice to focus on 1D CNNs was driven by their specific strengths in handling spatially structured data, which is central to the objectives of our research.

Other comments (mostly minor ones):

"Punctual": Should it not be "pointwise"  ? It seems that punctual has only the meaning as on-time (https://dictionary.cambridge.org/no/ordbok/engelsk/punctual )

Thank you for highlighting the terminology issue regarding the use of 'punctual'. Based on your valuable feedback, we agree that 'point-wise' is indeed the more appropriate term for our context. Consequently, we will revise our manuscript and replace all instances of 'punctual' with 'point-wise' to ensure accuracy and clarity.

Line 65: "This originates from the fact that MLPs are trained on individual data points and provide pointwise outputs, which makes the generation of regular profiles challenging as the NN does not take into account the vertical neighbors of predicted variables." This description is a bit too simple. This all depends on how the MLP is trained. One could (theoretically) also feed a profile and get a profile in return. For long profiles however this would be computational and prohibitive.

We acknowledge the reviewer's point that the architecture of MLPs and their training process can vary based on the approach taken. The specific MLP architectures used as a comparison in this paper (i.e. Fourier et al., 2020 and  Pietropolli et al 2023) are  trained on individual data points to provide point-wise output and only subsequently used to reconstruct profiles While we recognize that an MLP architecture capable of processing vector inputs and outputs could potentially be employed – as suggested by the reviewer – such a setup would significantly increase computational demands. Most importantly, a 1D convolutional architecture is characterized by better performance when handling vectorial input structures characterized by spatial correlation between different points in the vector. This choice was made based on existing literature and empirical evidence demonstrating the effectiveness of 1D convolutional architectures in analogous type of 1D data, such as processing of medical signals (Goodfellow et al 2018).

According to the suggestion of the reviewer, we will modify the sentence (in old line 65) in as follows:

"In fact, when these methods are used to  **predict** profiles from Argo float measurements, they may generate  irregularities in the reconstruction. **This is possibly due to the fact that these MLP uses point-wise data for input and output.** ."

Line 105: "we only selected profiles that were marked with quality flags (QFs) of 1, 2, or 8 for variables such as temperature, salinity, nitrate, and chlorophyll." Please also provide the labels associated with these classes.

The section 2 (presentation of the dataset) has been thoroughly revised also after comments from Rev#2. In particular, we will explain better the quality check used for the preparation of the training/testing/validation dataset. The meaning of the quality flags will be explained. The new text is as follows:

"The data used to train and test the architecture discussed in this paper comes from the  **BGC-Argo** program (Bittig et al. (2019)), specifically the Argo float collecting also biogeochemical variables.

**Our investigation used BGC-Argo S-profile data for the Mediterranean Sea downloaded from the Coriolis Argo GDAC (Argo, 2023; last visit on August 2022) and the analysis considered only Delayed Mode (DM) and Adjusted Real-Time Mode (RT) data for the period from 1-7-2013 to 31-12-2020 ensuring a larger number of DM data. DM data undergoes a more rigorous quality control process and is typically released a few months later to their sampling (Li et al., 2020).**

**Dataset was quality checked retrieving only complete profiles with Quality Flags 1 (good data), 2 (probably good data), 5 (value changed) and 8 (interpolated) for temperature, salinity, nitrate and chlorophyll. Furthermore, three specific quality check steps were applied for Bbp700 based on the study by Dall'Olmo (2022): missing-data test (profiles with substantial amount of missing data); high-deep-value test (profiles with unusually high bbp700 value at depth) and negative-bbp test (profiles with negative bbp700 values). Finally, for each of three variables (nitrate, chlorophyll and bbp700), profiles were only considered if the corresponding oxygen, salinity and temperature profiles were also available. As a result of the QC, the number of profiles for each variable used in the train, test and validation is reported in Table 3, the float spatial distribution is shown in Figure 2 and the dataset is available at the following repository (Amadio et al., 2023)."**

References used which will be added to bibliography:

- Amadio, C., TERUZZI, A., Feudale, L., BOLZON, G., DI BIAGIO, V., Lazzari, P., Álvarez, E., Coidessa, G., Salon, S., & COSSARINI, G. (2023). Mediterranean Quality checked BGC-Argo 2013-2022 dataset [Data set]. Zenodo. https://doi.org/10.5281/zenodo.10391759

Table 1: I think you should mention that 32 is the batch size in the caption. Some authors omit the batch size in such tables as it is considered an adjustable hyperparameter.

The 32 in Tables 1 and 2 refers to the minibatch size used during the training of our model. We understand the potential confusion this might cause. We will revise the two tables to remove the minibatch size.

Additionally, we note that the batch size, being a critical hyperparameter of the training process, is already detailed in Table 3. To maintain clarity and avoid redundancy, we will keep the information regarding the batch size contained in Table 3.

Line 133: "the sampling date (specifically year and day), geolocation, and geographic coordinates (latitude and longitude)": What is geolocation if not geographic coordinates ?

The term 'geolocation' indeed corresponds to 'geographic coordinates.' Based on this feedback, we will revise line 133 in our manuscript, as follow:

"the sampling date (specifically year and day),  and geographic coordinates (latitude and longitude)"

Line 121: "MLPs employed to transform punctual data into a vectorial shape - necessary for the training of the convolutional component." Why did you not consider a more obvious approach to simply repeated the sampling data and coordinates along the depth dimension. After all, all values in the profiles are considered to be at the same time and date. With the depth information that, the NN can specially on depth and thus come an entirely convolutional neural network.

The main motivation behind the use of a MLP for transforming scalar inputs into vectorial ones is that the same scalar value can have different impacts at different depths and this can be learned by the MLP performing the 1x1 to 1x200 transformation (we justify this choice in line 137 of the manuscript). In particular, while not widespread, a similar approach was already applied in "Learning Hand-Eye Coordination for Robotic Grasping with Deep Learning and Large-Scale Data Collection" by Levine et Al. (2016). In that paper the authors used a fully connected layer with 64 units whose outputs were added pointwise to 64 response maps (i.e., they provided the equivalent of a bias term). In our case we still employ fully connected layers but, instead of performing a pointwise addition to multiple response maps, we directly add the output as a channel. In both cases the idea of using a MLP to adapt the data before inserting it in the CNN architecture is present.

Reference used for the response:

- Levine, S., Pastor, P., Krizhevsky, A., Ibarz, J., & Quillen, D. (2018). Learning hand-eye coordination for robotic grasping with deep learning and large-scale data collection. *The International journal of robotics research*, *37*(4-5), 421-436.

Section 3.2 and table 2: The architecture of the convolutional layer is rather surprising. In a UNet or convolutional autoencoder one would expect to have first all the convolutional layers

(with a stride of 2 or with pooling layers) followed by all the deconvolution layers to get to the original depth dimension. In this paper they are mixed. For the convolutional layers you use kernel size of 2, 3 and 4. Can you explain why you use different sizes?

Our design choices were inspired by the study conducted by Goodfellow et al., 2018 which aimed to create a convolutional neural network (CNN) for ECG rhythm classification. We adopted different kernel sizes (2, 3, and 4) to manipulate the network's receptive field. By varying these kernel sizes, our goal was to capture features at different scales and depths within the data, an approach particularly relevant to our context where understanding the spatial correlation and complex features of the data is essential. The mix of convolution and deconvolution layers, diverging from the traditional sequential approach, was chosen to provide a more flexible representation of features. This methodology, inspired by the adaptability and effectiveness demonstrated in Goodfellow's study, is employed to enhance our network's capabilities in handling complex spatial data.

Line 175: "Second, to mitigate overfitting phenomena, a regularization term known as λ-regularization is employed, which penalizes complex curves in proportion to the square of the model's weights (Zou and Hastie (2005))" There is no square in equation (2). Can you clarify if you use L1 or L2 regularization?

We thank the reviewer for highlighting the inconsistency related to the regularization term. Indeed, the regularization method we employed is L2 regularization, which penalizes the model in proportion to the square of its weights. We have accordingly revised Equation (2) .

Equation 2: I think you forgot the alpha coefficient in this equation. Also consider to use a different symbol as alpha is already used for something different in equation (1).

We appreciate the reviewer's observation regarding the omission of the coefficient in Equation (2). To rectify this, we will include the missing coefficient in Equation (2). Additionally, to avoid any confusion with the symbol used in Equation (1), we will choose a different symbol for the coefficient both in the equation and the main text.

193: "validation sets were chosen as 80, 10, and 10." Add %

Thank you for pointing out the omission of the percentage symbol in our description of the dataset division. We will update the text to to correctly reflect the distribution as:

"80**%**, 10**%**, and 10**% of the total number of measurements**".

Line 203: "Adadelta (Zeiler (2012)) is the algorithm that is selected as the optimizer for training the network due to its ability to dynamically adapt over time using only first-order information": Can you be more specific what "first-order information" means here?

In the context of optimization algorithms like Adadelta, "first-order information" refers to the use of first-order derivatives or gradients of the objective function (often the loss function in machine learning). This is in contrast to second-order methods that use second-order derivatives or curvature information. In order to render the sentence more clear, we will modify it as follows:

"Adadelta (Zeiler (2012)) is the algorithm that is selected as the optimizer for training the network due to its ability to dynamically adapt over time using only first-order  **derivatives of the objective function**".

Table 3: please replace $1e^-7$ by $10^{-7}$. In equation $e$ means the Euler number and $e^{-7}$ would be $exp(-7)$ (also on Table 6, 7,8).

Thank you for pointing out the notation error in Tables 3, 6, 7, and 8. We understand the reviewer's concern regarding the use of '$1e^-7$' and the potential confusion with Euler's number, e. Based on your suggestion, we will correct this notation to '$10^{-7}$' in all the mentioned tables.

Line 346: "On the other hand, we have demonstrated that the RMSE for the PPCon architecture is 0.61" Which unit are you using?

Thank you for highlighting the omission of the unit in Line 346. We indeed forgot to specify the unit for the RMSE value of the PPCon architecture. Moreover, we did not mention which variable we were referring to. The considered variable is nitrate and thus the unit used for RMSE is µmol/kg. We will update the text to include this unit, as follow:

"On the other hand, we have demonstrated that**, for example,** the RMSE for the PPCon architecture **regarding the prediction of nitrate is 0.52 µmol/kg**".

Line 27: "These instruments are essential to advance our knowledge of the biogeochemical state of the ocean, as one of their principal advantages is the assimilation into ocean biogeochemical models"Advantages -> use cases?

We agree with the reviewer's suggestion regarding the terminology: the term 'use cases' indeed provides a more specific and practical perspective compared to 'advantages.' We will modify the sentence as follow:

"These instruments are essential to advance our knowledge of the biogeochemical state of the ocean, as one of their principal  **use cases** is the assimilation into ocean biogeochemical models".

References: Please add a DOI where one is available (see https://www.geoscientific-model-development.net/submission.html) https://doi.org/https://doi.org/10.17882/42182, 2000 -> , https://doi.org/10.17882/42182, 2000

Thanks for spotting these lacks :  we will add the DOIs where necessary in the revised manuscript.

In general, replace Adadelta (Zeiler (2012)) by Adadelta (Zeiler, 2012) and similar. (this is \citep in latex).

We thank the reviewer for pointing out this text format issue, we will replace the format of the reference according to this suggestion in the revised text.

References used in the response:

- Goodfellow, I., Bengio, Y., & Courville, A. (2016). Deep Learning. MIT Press.
- Bai, S., Kolter, J. Z., & Koltun, V. (2018). An empirical evaluation of generic convolutional and recurrent networks for sequence modeling. arXiv preprint arXiv:1803.01271.
- Kiranyaz, S., Ince, T., & Gabbouj, M. (2021). 1D Convolutional Neural Networks and Applications: A Survey. Mechanical Systems and Signal Processing, 151, 107398.
- Goodfellow, Sebastian D., et al. "Towards understanding ECG rhythm classification using convolutional neural networks and attention mappings." *Machine learning for healthcare conference*. PMLR, 2018.
- doi.org/10.5281/zenodo.10391759. Amadio, C., TERUZZI, A., Feudale, L., BOLZON, G., DI BIAGIO, V., Lazzari, P., Álvarez, E., Coidessa, G., Salon, S., & COSSARINI, G. (2023). Mediterranean Quality checked BGC-Argo 2013-2022 dataset [Data set]. Zenodo. https://doi.org/10.5281/zenodo.10391759

**Reviewer 2**

Rev#2.01 The authors describe a method that predicts nitrate, chla and bbp vertical profiles from geolocated and dated vertical profiles of temperature, salinity, and oxygen. With the help of CNNs, they use vertical profiles (data values and shape) rather than point-wise data values for these predictions. The method is trained and evaluated on BGC-Argo profiling float data acquired in the Mediterranean Sea.

We appreciate the constructive comments and suggestions from the Reviewer. We present our point-by-point responses to the Reviewer's comments below. The Reviewer's comments are in blue, our responses follow each comment in black. In each response, we detail the changes we propose to make to the manuscript in order to address the following comments, and include the proposed modified text and/or figure (in red).

Rev#2.02 What is very interesting in this work is that -- in contrast to previous work -- they propose to take a stronger advantage of the data context by using a convolutional neural network together with entire profiles for prediction. This is novel and innovative. What requires attention and significant improvement, however, is the presentation and clarity of their work. The manuscript starts off with a well-detailed and well-written introduction, but attention to detail and readability suffer the further one goes into the later sections. Unfortunately to an extent, at which it in places becomes unclear to a reader what is meant by the authors, e.g.:

l.345 "BGC-Argo float data for oxygen, nitrate, and chlorophyll concentrations exhibit RMSE values evaluated at 5.1±0.8µmol/kg, 0.25±0.07µmol/kg, and 0.03±0.01mg/m3, respectively. On the other hand, we have demonstrated that the RMSE for the PPCon architecture is 0.61" -- ??? What's the 0.61 related to? No unit, no oxygen/nitrate/chla given.

Thank you for highlighting the omission of the unit in Line 346. We indeed forgot to specify the unit for the RMSE value of the PPCon architecture. Moreover, we did not mention which variable we were referring to. The considered variable is nitrate and thus the unit used for RMSE is µmol/kg. We have now updated the text to include this unit, as follow: "On the other hand, we have demonstrated that, **for example,** the RMSE for the PPCon architecture **regarding the prediction of nitrate is 0.52 µmol/kg**".

Rev#2.03 This is unfortunate and definitely needs attention before a re-submission. Starting from Section 3 and onwards, a thorough proof-read and maybe re-write would be recommended.I I take it that this manuscript is part of a thesis, with later parts likely written in a rush. This is very understandable, so I would like the authors to take my comments as advise on how to get their excellent idea into a shape that mirrors it's worth.

We thank the reviewer for the encouraging feedback. We have read all the following comments, and we will make all necessary adjustments in order to have the discussion more clear and precise. We will propose changes and re-written parts following general comments raised by Rev#2. Subsequently, the entire manuscript will be thoroughly read and checked for consistency when our responses to the reviewers' comments are accepted.

*General comments*:

- Rev#2.04 While the method can be applied anywhere, the present study focuses on the Mediterranean Sea. This must be mentioned somewhere prominently/early, e.g., either in the title or abstract. A reader cannot be left searching for the regional coverage until somewhere late in section 4, if one happens to glance over the one single sentence in l. 105. With almost all previous text (Intro, Argo description, etc.) referring to the global system.

We present the PPCon as a general method that can be applied anywhere, and we demonstrate its validity/effectiveness using the case study of the Mediterranean Sea. We believe that the Mediterranean Sea is an ideal case for testing new approaches: the density of profiles is fortunately high and the BGC-Argo profiles show sufficient (temporal and spatial) variability to cover a wide range of marine biogeochemical regimes (e.g. from mid-eutrophic to ultra oligotrophic conditions). The choice of the Mediterranean Sea will be mentioned prominently and early in the Abstract and in the introduction. However, we would prefer to leave the title as it is, as the aim of the paper is to present and make available a general method and its Python code.

We propose the following modifications to the manuscripts:

1) in abstract at old lines 10-16:
"In this study, we present a novel one-dimensional Convolutional Neural Network (1D CNN) model to predict profiles leveraging the typical shape of vertical profiles of a variable as a prior constraint during training. In particular, the Predict Profiles Convolutional (PPCon) model predicts nitrate, chlorophyll and backscattering (bbp700) starting from the date, geolocation, and temperature, salinity, and oxygen profiles. Its effectiveness is demonstrated using a robust BGC-Argo dataset collected in the Mediterranean Sea for training and validation. Results, which include quantitative metrics and visual representations, prove the capability of PPCon to produce smooth and accurate profile predictions overcoming some limitations of previous MLP applications."

2) in the introduction section, the old lines 78 and 79 will be replace with the following paragraph:
"PPCon approach is tested with a robust Argo dataset collected in the Mediterranean Sea. The Mediterranean Sea, a semi enclosed marginal sea, presents a substantial high density of BGC-Argo profiles thanks to dedicated programs such as ARGO-Italy and French NAOS initiative (D'Ortenzio et al., 2020). This particularly fortunate situation has already made the Mediterranean a successful case for the development of biogeochemical modelling approaches based on BGC-Argo. For example, BGC-Argo is being integrated into the biogeochemical prediction model of the Mediterranean component of the Marine Copernicus Service (Cossarini et al., 2019; Teruzzi et al., 2021; Coppini et al., 2023)."

References used which will be added to bibliography:

- D'ortenzio, F., Taillandier, V., Claustre, H., Prieur, L. M., Leymarie, E., Mignot, A., ... & Schmechtig, C. M. (2020). Biogeochemical Argo: The test case of the NAOS Mediterranean array. *Frontiers in Marine Science*, 7, 120.

- Coppini, G., Clementi, E., Cossarini, G., Salon, S., Korres, G., Ravdas, M., ... & Zacharioudaki, A. (2023). The Mediterranean Forecasting System–Part 1: Evolution and performance. *Ocean Science*, *19*(5), 1483-1516.
- Teruzzi, A., Bolzon, G., Feudale, L., & Cossarini, G. (2021). Deep chlorophyll maximum and nutricline in the Mediterranean Sea: emerging properties from a multi-platform assimilated biogeochemical model experiment. *Biogeosciences*, *18*(23), 6147-6166.
- Cossarini, G., Mariotti, L., Feudale, L., Mignot, A., Salon, S., Taillandier, V., ... & d'Ortenzio, F. (2019). Towards operational 3D-Var assimilation of chlorophyll Biogeochemical-Argo float data into a biogeochemical model of the Mediterranean Sea. *Ocean Modelling*, *133*, 112-128.

- Rev#2.05 In general, the manuscript would benefit from more clarity (one example: Make sure to use the same term for the same thing). And from more specifics throughout the text (one example: "nitrate, chla, and bbp" instead of "all inferred variables"; Does "output variable" in l.124 refer to output of the MLP or output of PPCon?). If you can name it, then name it and do not find an alternative description.

Many thanks for the comments. Variables will be named nitrate, chlorophyll and particulate backscattering coefficient at 700nm. Variables will be abbreviated in NO3, CHL and bbp700 when appropriate.

Finally, the sentence at old line 124 will be modified as follows:

"The input variables for PPCon include sampling data, geolocation, temperature, salinity, and oxygen, while the **PPCon** output comprise vertical profiles for nitrate, chlorophyll, and BBP."

- Rev#2.06 Who is your target audience with this paper? I believe it's an oceanographic community? For this, it is surprising to not see a map of any sort, which would help a reader to follow along (and to put it into his/her oceanographic background context).
I'd therefore like to see a map to show all BGC-Argo float profile positions used for algorithm training, testing, and (independent) validation (e.g., each with a different colour), to show the specific float positions relating to the selected profile plots (e.g., with a different marker), and to show the different areas for the posterior analysis (e.g., as boxes). If you're limited in number of figures/tables, drop one of the existing ones (or combine figures into one), because a map is much more important than any (anecdotal) illustration.

Many thanks for the comment. We agree that the PPCon method and code should target the oceanographic community. Thus, the paper will benefit from a more friendly presentation of the data used for training, testing, validation and external validation. We propose to include a new figure reporting the position of the float profiles for the three variables and the limits of the sub-regions used for the statistics computation.

[Figure]

"New Figure 2: position of BGC-Argo float profiles for bbp700 (red), chlorophyll (blue) and nitrate (green) that also have oxygen data. BGC-Argo floats used for external validation (black and labels). Geographical limits of sub-regions (dashed boxes): North Western Mediterranean (NWM), South Western Mediterranean (SWM), Tyrrhenian (TYR), Ionian and southern Adriatic Sea (ION) and Levantine (LEV). Position of the 4 BGC-Argo float profiles used for the external validation (black and numeric labels)."

The description of the new figure will be included in the re-written section 2 (dataset presentation). Changes to section 2 are reported in the next comments (e.g. Rev#2.07). Additionally, reference to the new figure 2 will be added appropriately in sections 3 and 4 and the old Table 4 (geographical limits of sub-regions) will be removed.

- Rev#2.07 Argo (and BGC-Argo) is a living dataset, which constantly evolves both with new data being added but also with existing data being re-evaluated/newly adjusted. Which means that, e.g., today's 2016-sampled profile (even if in delayed mode) may look different from when you downloaded the same profile last year or from when you will download it in 1 year's time (https://argo.ucsd.edu/data/data-faq/#reD). It is therefore important to make one's own work traceable, e.g., by stating the date one downloaded the dataset and from which source (e.g., the GDAC). Even better would be, e.g., use of one of the monthly snapshots of the Argo GDAC doi, which specifically refer to the state of the Argo dataset at the given time (https://argo.ucsd.edu/data/data-faq/#DOI).

Many thanks for the comment, we were not aware of the snapshot DOI. OGS is part of the Mediterranean Modelling Center of the Marine Copernicus Service (Coppini et al., 2023) and BGC-Argo are used operationally for data assimilation in the biogeochemical model. That is to say that we are updating our BGC-Argo dataset every day with the new profiles and the updated profiles. The analysis presented in this paper was done using a dataset updated in August 2022 and considering only profiles till 31-12-2020 to have a larger number of DM

data. To clarify the origin of the dataset, we will rewrite section 2 including a reference to the BGC-Argo doi: http://doi.org/10.17882/42182 and the date of the last visit.

Taking into account the following comments and one of Rev#1, the old lines 100-112 will be rewritten as follows:

**"Our investigation used BGC-Argo S-profile data for the Mediterranean Sea downloaded from the Coriolis Argo GDAC (Argo, 2023; last visit on August 2022) and the analysis considered only Delayed Mode (DM) and Adjusted Real-Time Mode (RT) data for the period from 1-7-2013 to 31-12-2020 ensuring a larger number of DM data. DM data undergoes a more rigorous quality control process and is typically released a few months later to their sampling (Li et al., 2020).**

**Dataset was quality checked retrieving only complete profiles with Quality Flags 1 (good data), 2 (probably good data), 5 (value changed) and 8 (interpolated) for temperature, salinity, nitrate and chlorophyll. Furthermore, three specific quality check steps were applied for Bbp700 based on the study by Dall'Olmo (2022): missing-data test (profiles with substantial amount of missing data); high-deep-value test (profiles with unusually high bbp700 value at depth) and negative-bbp test (profiles with negative bbp700 values). Finally, for each of three variables (nitrate, chlorophyll and bbp700), profiles were only considered if the corresponding oxygen, salinity and temperature profiles were also available. As a result of the QC, the number of profiles for each variable used in the train, test and validation is reported in Table 3, the float spatial distribution is shown in Figure 2 and the dataset is available at the following repository (Amadio et al., 2023)."**

References used which will be added to bibliography:

- Argo (2023). Argo float data and metadata from Global Data Assembly Centre (Argo GDAC). SEANOE. https://doi.org/10.17882/42182; last visit on August 2022
- Amadio, C., TERUZZI, A., Feudale, L., BOLZON, G., DI BIAGIO, V., Lazzari, P., Álvarez, E., Coidessa, G., Salon, S., & COSSARINI, G. (2023). Mediterranean Quality checked BGC-Argo 2013-2022 dataset [Data set]. Zenodo. https://doi.org/10.5281/zenodo.10391759

- Rev#2.08 Figure captions should provide a text description of what is presented so that their content can be understood without the main text (applies, e.g., to Figure 1, but all figures in general). Think of a lazy reader, who wants to get the main points of your paper just by looking through your figures. At least what is presented in which colour/marking and against what must become clear (e.g., Fig. A1: continuous vs. dashed lines??; Fig. 2: Which MLP is shown?).

According with the suggestions of all reviewers, we propose the following more detailed version of the paragraph (in bold the new text):

"Figure 1: Illustration of the principal architectural components of the PPCon model: i) **MLP network to transform the point-wise inputs (day, year, latitude, and longitude) into vectorial form; ii) vectorial inputs (profiles of temperature, salinity, and oxygen and**

**output of the MLP); iii) structure of encoder-decoder of a 1D CNN architecture; iv) output vector representing the vertical profile of one of the target variables (nitrate, chlorophyll, or backscattering)."**

**"New Figure 3 (old Figure 2):** Profiles of nitrate for some selected floats (WMO numbers **and cycle** in the title) and dates. **Profile dates and geolocations are reported in Table 5.** Comparison between measured profile (green lines),  and PPCon reconstruction (blue dash-dotted lines). Profiles are from the subset used for the test."

**"New Figure 4 (old Figure 3)**: Profiles of chlorophyll for some selected floats (WMO numbers **and cycle** in the title) and dates. **Profile dates and geolocations are reported in Table 5.** Comparison between measured profile (green lines) and PPCon reconstruction (blue dashed lines). Profiles are from the subset used for the test."

**"New Figure 5 (old Figure 4):** Profiles of bbp700 for some selected floats (WMO numbers **and cycle** in the title) and dates. **Profile dates and geolocations are reported in Table 5.** Comparison between measured profile (green lines) and PPCon reconstruction (blue dashed lines). Profiles are from the subset used for the test."

**"New Figure 6 (old Figure 5)**: Plot of the RMSE distribution with respect to the data variability **(on the x-axis)** and training dataset size **(on the y-axis)**. Different sub-areas **are represented by different** colour of the symbol, and different seasons **are represented by different** symbol fill pattern. RMSE values are categorized by the size of the symbols, **and bigger symbols correspond to bigger RMSE values**."

"Figure A1: Comparison of the mean of PPCon predicted profiles with the mean of sampled values measured by the float instruments in the test set, for all the variables investigated. **Results are divided among different geographic areas, the dashed lines represent sampled values, while continuous lines represent PPCOn predictions."**

- Rev#2.09 Check/reconsider the number of significant digits given for statistical metrics throughout the text, and make them coherent (and not more 'accurate' than realistic). (E.g., on the RMSE of nitrate, ... in tables 6-8)

In order to provide a more coherent and realistic statistical analysis of the results obtained with the PPCon model, we will modify the number of significant digits for nitrate, chlorophyll and bbp700 in Table 6, Table 7 and Table 8. Specifically, we will adopt two significant digits for all those variables (eg 0.655 will become 0.66).

*Section by section comments*: (focus on style/structure)

- Rev#2.10 The introduction is extensive and well-written with appropriate referencing (though referencing around global programs are a bit Med Sea centric; which is probably fine once the regional scope becomes clear).

We appreciate the reviewer's positive comments regarding the introduction. Taking into account also one of the previous comments of the reviewers and as mentioned in our earlier

response (Rev#2.04), we have taken steps to clarify the regional scope of our study in the introduction.

- Rev#2.11 For the scope of the present paper, section 2 Dataset can be drastically reduced: Lines 87-104 can be condensed into two sentences: "We used data from the evolving BGC-Argo network, which uses profiling floats ... ." and "BGC-Argo data were accessed from the Argo GDAC." Then add which data mode you used (only delayed-mode adjusted data? Or also real-time adjusted data? Hopefully no unadjusted real-time data?), what kind of files (likely the s- rather than b-profiles?), and, especially important, add how many profiles you had in your data selection prior and past QC/preprocessing.
The size of the training dataset remains unclear to me in the present manuscript.

We agree that section 2 can be revised and the used dataset better presented. Taking into consideration. Taking into account one of the previous comments and one of Rev#1 comment, section 2 will be revised as follows:

"The data used to train and test the architecture discussed in this paper comes from the  **BGC-Argo** program (Bittig et al. (2019)), specifically the Argo float collecting also biogeochemical variables.

**Our investigation used BGC-Argo S-profile data for the Mediterranean Sea downloaded from the Coriolis Argo GDAC (Argo, 2023; last visit on August 2022) and the analysis considered only Delayed Mode (DM) and Adjusted Real-Time Mode (RT) data for the period from 1-7-2013 to 31-12-2020 ensuring a larger number of DM data. DM data undergoes a more rigorous quality control process and is typically released a few months later to their sampling (Li et al., 2020).**

**Dataset was quality checked retrieving only complete profiles with Quality Flags 1 (good data), 2 (probably good data), 5 (value changed) and 8 (interpolated) for temperature, salinity, nitrate and chlorophyll. Furthermore, three specific quality check steps were applied for Bbp700 based on the study by Dall'Olmo (2022): missing-data test (profiles with substantial amount of missing data); high-deep-value test (profiles with unusually high bbp700 value at depth) and negative-bbp test (profiles with negative bbp700 values). Finally, for each of three variables (nitrate, chlorophyll and bbp700), profiles were only considered if the corresponding oxygen, salinity and temperature profiles were also available. As a result of the QC, the number of profiles for each variable used in the train, test and validation is reported in Table 3, the float spatial distribution is shown in Figure 2 and the dataset is available at the following repository (Amadio et al., 2023)."**

Regarding the QC/preprocessing data, we decided to publish the dataset used for training, testing and validation in a dedicated Zenodo repository (Amadio et al., 2023) to allow full reproducibility of the PPCon results. The repository of Amadio et al. (2023) contains some additional details about the QC/preprocessing used by the Marine Copernicus Mediterranean Modelling Centre in OGS. In particular, preprocessing is presented in the Coppini et al. (2023) publication, and the QC/preprocessing code (named bit.sea) is available in a dedicated Zenodo repository (Bolzon et al., 2023). To avoid including too much

information in Table 3, we would prefer to leave Table 3 as it is now, unless requested by the reviewer.

The sizes of the training, testing, and validation sets correspond to the percentages reported in the main text: 80% of the data is used for training, 10% for testing, and 10% for validation. The sentence at old line 193 will be rewritten as follows:

"The sizes of the training, testing, and validation sets were chosen **as 80%, 10%, and 10% of the total number of profiles (Table 3)"**

References used for the response:

- Coppini, G., Clementi, E., Cossarini, G., Salon, S., Korres, G., Ravdas, M., ... & Zacharioudaki, A. (2023). The Mediterranean Forecasting System–Part 1: Evolution and performance. *Ocean Science*, *19*(5), 1483-1516.
- Amadio, C., TERUZZI, A., Feudale, L., BOLZON, G., DI BIAGIO, V., Lazzari, P., Álvarez, E., Coidessa, G., Salon, S., & COSSARINI, G. (2023). Mediterranean Quality checked BGC-Argo 2013-2022 dataset [Data set]. Zenodo. https://doi.org/10.5281/zenodo.10391759
- Bolzon G., Teruzzi A., Salon S, Di Biagio V., Feudale F., Amadio C., Coidessa G., & Cossarini G. (2023). bit.sea (1.7). Zenodo. https://doi.org/10.5281/zenodo.8283692

- Rev#2.12 section 3 PPCon: The first paragraph is a repetition of the introduction. It needs to be there only once and I would recommend it to be only in the introduction.

We agree with the reviewer's observation regarding the repetition of content in the first paragraph of Section 3, which mirrors information already presented in the introduction. In order to avoid redundancy, we will remove the first paragraph from Section 3.

- Rev#2.13 I assume the "32" in tables 1 and 2 is the minibatch size? As such, it shouldn't be listed as part of the layer size, which I find very confusing. (Consider the prediction step of one profile - no batch size is relevant here.)

The 32 in Tables 1 and 2 refers to the minibatch size used during the training of our model. We understand the potential confusion this might cause. We will revise the two tables to remove the minibatch size.

Additionally, we note that the batch size, being a critical hyperparameter of the training process, is already detailed in Table 3. To maintain clarity and avoid redundancy, we will keep the information regarding the batch size contained in Table 3.

- Rev#2.14 section 4.1 is again rather general (except for one paragraph), and I wonder whether the majority of its content should rather go to the introduction.

We appreciate the reviewer's feedback on Section 4.1.
While it is indeed somewhat general, we included a broader and more general context regarding the neural network's training procedure to ensure clarity. Since not all readers are familiar with concepts such as mini-batching or network optimizers, we think that it is

important to include these details in the experimental study section to ensure comprehensive understanding and facilitate the complete reproducibility of the PPCon architecture.

In order balance the generality and the necessary details of this part of the discussion, we will reformulate the paragraph 4.1 as follows:

"We divided the dataset into three subsets: training, testing, and validation. The training set was used for model training and parameter optimization. The testing set was utilized to evaluate the model's performance on unseen data and assess its generalization ability. Finally, the validation set was employed for hyperparameter tuning and model selection. The dataset was randomly partitioned, ensuring that each subset contained a representative distribution of the overall data characteristics. The sizes of the training, testing, and validation sets were chosen **as 80%, 10%, and 10% of the total number of profiles (Table 3).**  Moreover, before operating this partition, a few float instruments have been selected and all of their measurements have been excluded from both the training, test, and validation set. These samples will be used as an external validation dataset. The metrics and the performances over this external validation dataset are a more effective indicator of the generalization capabilities of the PPCon model with respect 205 to the metrics on the test set.

To train the NN model efficiently, the input dataset is partitioned into minibatches, where each minibatch contains 32 samples. The batch size **is a hyperparameter** which determines the number of samples processed before updating the model weights . By processing multiple samples in a minibatch, the model can update its parameters more frequently, which can lead to faster convergence and improved generalization performance (Bottou (2010)).

Adadelta (Zeiler (2012)) is the algorithm that is selected as the optimizer for training the network due to its ability to dynamically adapt over time using only first-order  **derivatives of the objective function**. This method eliminates the need for manual tuning of the learning rate and has been found to exhibit robustness.

It is worth recalling that the PPCon architecture includes a 1D CNN and four MLPs, which convert point-wise input into a vector form suitable for use by the CNN. The MLPs and the CNN component of PPCon were trained using the same optimizer, with concurrent weight updates across all networks. This approach enables the joint learning of optimal information transfer from point-wise input to vector form, as well as the accurate generation of predicted profiles based on the input tensor. To accelerate the training process, the model was trained using a GPU (graphics processing unit), which allowed for parallelized computation of the forward and backward passes. The model's performance was evaluated once every 25  epoch by assessing its ability to predict outcomes on the test set, which consists of previously unseen data. To prevent overfitting and minimize computational burden, we introduced an early stopping routine. Specifically, the training was interrupted if the error metrics on the validation set increased for two consecutive evaluations (i.e. after 50 epochs of training). The final model selected was the one trained before the two 225 consecutive test loss increases."

- Rev#2.15 Section 4.3 would largely benefit from a map, e.g., to illustrate the "uneven spatial and temporal distribution of the profiles" (l. 254) which remains elusive otherwise.

Regarding Section 4.3, we have taken the reviewer's suggestion to include a map illustrating the uneven spatial distribution of the profiles in the Mediterranean Sea (see our reply to one of the previous comments).

The section will be revised to include a detailed explanation of the map (which is the new Figure 2, reported in Rev#2.06), ensuring that the distribution of the floats across various Mediterranean Sea areas is clearly described. Section 4.3 will be modified as follow:

**"4.3 A posterior validation analysis of PPCon**

To validate the PPCon architecture, we conducted a thorough analysis of its  performance in  **five** geographic areas **(NWM, TYR, SWM, ION and LEV in New Figure 2)** and across the  **four** seasons**: winter (JFM), spring (AMJ), summer (JAS) and autumn (OND). While the PPCon model is trained on the entire dataset, this subdivision is only used to analyze the performance retrospectively to check whether the non-uniform spatial distribution of the profiles and the natural variability of the profiles (e.g. depth and slope of the nitracline, or depth and intensity of the DCM) have an influence. In particular, the RMSE is calculated for the reconstructed profiles in each area and season to verify the presence of any bias in the accuracy of PPCon in capturing the spatial and temporal variability in the Mediterranean Sea.**

~~The choice underlying this investigation originates from the fact that diverse geographic areas and seasons are known to have distinct profile properties, such as the shape of the vertical profiles (e.g. depth and slope of the nitracline, or depth and intensity of DCM) and the values at the surface and in deep water. Our goal is to evaluate the model's ability to accurately capture these variations. We wish to once again note the model is trained on the entire dataset, and this division is purely for a posteriori analysis of the performances. This a posteriori analysis of the performances has the objective of identifying possible influence and bias of the uneven spatial and temporal distribution of the profiles on the performance of the PPcon model.~~

- Rev#2.16 Section 5 is more clearly written again, which is good!

Rev#2.17 A question I had while reading the RMSE values/patterns discussion (3rd paragraph) is how this relates to the range of values and variability of nitrate/chla/bbp, e.g., within a given season. Eventually, this information is touched upon, but I'd suggest to move the RMSE discussion, its temporal evolution and patterns (third paragraph) more closely to the last paragraph of part 5.0.

We acknowledge the reviewer's suggestion regarding the organization of Section 5.
In light of this feedback, we will revisit the order of the paragraphs and revise them accordingly.

Rev#2.18 This last paragraph (of section 5.0) and discussion of Figure 5 contains a lot of (valuable) information! I would encourage the authors to spend more time/space on its

discussion, so that it can be adequately digested by a reader, and I would like to see this presented more extensively to get better context, e.g., to the RMSE/performance vs. season vs. natural variability.

We agree with the reviewer, and propose the following expanded discussion of Figure 5. We report the new Figure 5 and the relative improved discussion:

[Figure]

"In order to understand the impact of the training set numerosity and of the variability of profiles on the quality of the PPCon predictions we investigated the relation between these quantities and the PPCon error. Specifically,  **Figure 5 illustrates the RMSE values computed for the reconstructed profiles subdivided in five geographic areas and four seasons. RMSE values, which are indicated by the size of the symbols, are plotted against the variability of the training set (quantified by the standard deviation on the x-axis) and the size of the training set (on the y-axis)**. This figure offers also valuable insight into the geographical and seasonal distribution of the training dataset dimension.

**A general observation from these plots is that the South Western Mediterranean exhibits the smallest number of training samples, while the largest numbers are in the North Western Mediterranean (NWM) and Levantine areas. Natural variability changes across sub-basins with higher values of standard deviations in the western sub-basins (i.e., SWM, NWM and TYR). Variability and sample size show roughly homogeneous distribution among seasons. .**

The analysis of the nitrate plot reveals fairly homogeneous errors across natural variability and training sample size . **The North Western Mediterranean is the basin predicted with the lowest accuracy, while the South Western Mediterranean and Ionian have in general the lowest errors. In terms of seasonal variation, the RMSE values appear slightly lower during summer compared to winter and spring.**

In terms of chlorophyll and bbp700, both variables exhibit similar behavior. In particular, data availability appears to have no significant impact on the error, whereas RMSE tends to increase proportionally with the variability.

**Regarding the chlorophyll, the performances of the western sub-basins (i.e., NWM, SWM and TYR) are lower than the eastern sub-basins (LEV and ION), likely due to higher profile variability. Winter and autumn are the seasons with lower RMSE, while the highest error is predicted in Spring.**

**Regarding bbp700, better performances are observed in the Levantine and in the Ionian compared to the western sub-basins, probably due to their lower variability. Interestingly, summer and autumn performances are almost 50% better than winter and spring ones despite natural variability and sample size do not show appreciable differences among the seasons.**

- Rev#2.19 Section 6 is logically structured, but the content needs careful inspection so that it contains all information intended/required to be transmitted. (E.g., units/which parameter in l. 346; l. 371: Must add "in the Med Sea" to have this sentence work; ...).

Many thanks for the suggestion, we will revise appropriately the sentences of the section 6 to provide all information required.

In particular, changes will be as follows:

at old lines 335-337: "However, while MLP architectures **have demonstrated to**  provide good training and test errors **for point-wise input/output**(Pietropolli et al. (2023a); Fourrier et al. (2020); Bittig et al. (2018); Sauz.de et al. (2017)), **they can**  exhibit higher errors when predicting BGC-Argo profiles (Pietropolli et al. (2023a)**, Appendix B**)."

at old lines 334-336: "According to the analysis conducted by Mignot et al. (2019), the BGC-Argo float data for  nitrate and chlorophyll  exhibit RMSE values evaluated at  0.25±0.07µmol/kg, and 0.03±0.01mg/m3, respectively. On the other hand, **PPCon architecture produced BGC-Argo profile reconstruction with**  RMSE **values of**  0.52 **mmol/m3 and 0.08 mg/m3, for nitrate and chlorophyll, respectively."**

at old lines 350-358: "Although both MLPs and PPCon employ similar input information (date, geolocation, temperature, oxygen, and salinity), their treatment of this data differs significantly. **While the current MLP applications** process the input and output **as point-wise data** , PPCon utilizes vector representations of the vertical profiles. This approach  effectively exploit**s** the potential of a 1D CNN, which intrinsically preserves the characteristic profile shape of the input and output variables Kiranyaz et al. (2021). When comparing the predictive performance of these techniques in generating vertical profiles from float data, distinct differences emerge. MLPs **can**  produce profiles **affected by**  artificial discontinuity , while the profiles generated by PPCon exhibit a smoother and more realistic appearance (**Appendix B2**). **Additionally**,  **the RMSE values computed on the reconstructed nitrate profiles of the test sub-set confirms the better performance of the 1D CNN approach with respect to a MLP approach trained on point-wise data** (**Appendix B2**). "

at old lines 369-373: "Our PPCon architecture presents a valuable approach to harness the potential of the Argo and BGC-Argo network by enabling the synthetic generation of essential variables (chlorophyll, nitrate, and bbp700) even when these costly sensors are not present in the deployed floats. **For instance, the application of PPCon on Argo and oxygen profiles in the Mediterranean Sea for the period**  from 2013 to 2020 enabled the generation of **5234 (nitrate), 3879 (chlorophyll), 3307 (bbp700)** synthetic  profiles,  which means doubling the chlorophyll and bbp700 BGC-Argo profiles and more than tripling those of nitrate."

- The conclusion, again, is well-written, concise, and clear.

Specifics: (focus on content)

Rev#2.18 - l. 10: "resulting in irregularities such as jumps and gaps when used for the prediction of vertical profiles".
I would challenge this and think that neither of this is true. For well-trained MLPs or neural networks in general, regularization causes them to give smooth outputs in general. Jumpy behaviour is only to be seen in case of overfitting, i.e., where an operator chose to fit the training/testing data set too closely (for apparently good performance statistics) but neglected the regularization term, so that the trained network does not generalize sufficiently (and therefore gives seemingly erratic/jumpy predictions). But this is not sth. to blame the MLP architecture for, but to blame the network training/operator implementation. It seems that this claim is mostly supported by citation of the same authors' work (Pietropolli et al., 2023). Given the way it is presented and that it should stand for MLPs in general, it would largely benefit (or rather require!) support by other people's/lab's work.

Many thanks for the comment. We revised the results on the benefit of the Convolutional architecture over point-wise trained MLPs to reconstruct profiles. In particular, we propose to:

- compare PPCon with other 2 published MLP architectures specifically for the Mediterranean Sea: Fourier et al., 2020 and Pietropolli et al., 2023;

- use nitrate since it is the only variable common to all published architectures;

- use a common dataset (i.e., the subset of the test dataset) to perform a fair comparison among the different architectures.

The new results will be shown in a new Appendix and commented in the discussion section. Please refer to the point Rev#2.21 for the new Appendix B and modifications to the main text.

- Rev#2.19 Same comment on l.64-65. An MLP/point-wise prediction does not take into account neighboring data during prediction, true. -- But they do so during training due to their natural proximity in data state space and the MLP regularization. I.e., input data that are close together in data space (like from a single profile) do get output predictions that are

smoothly transitioning from one to the next output value (if adequately regularized/without overfitting). Nonetheless, CNNs can (likely) take advantage of neighboring data also during prediction, so it's worth to study. (This provides sufficient motivation for CNNs from their potential for improvement; there is no need to claim negative aspects on MLPs/current methods beyond this for motivation.)

We agree that our statement about MLPs not considering neighboring data during prediction might imply a limitation that is not entirely accurate.
It is indeed true that during training, MLPs can learn from data that are closely situated in the data space, leading to output predictions that transition from one value to the next.
In light of this, we will remove the sentence in question that suggests MLPs do not take into account neighboring data during prediction.
This change is intended to eliminate any potential confusion and to present a more accurate depiction of MLP capabilities.

we will revise the old lines 60-65 as follows:
"In fact, when these methods are used to  **predict** profiles from Argo float measurements, they may generate  irregularities in the reconstruction. **This is possibly due to the fact that these MLP uses point-wise data for input and output.**  "

- Rev#2.20 Continued: The authors write that (l. 335f) "MLP architectures can provide good training and test errors" (as by Pietropolli and 3 more references for MLPs) while "they have been found to exhibit higher errors when predicting BGC-Argo profiles" (only by Pietropolli but none of the 3 other references for MLPs) -- This should make someone a bit suspicious and check double if this holds in general (the way it is presented here).

As anticipated in point Rev#2.18, we propose to:

- compare PPCon with other 2 published MLP architectures specifically for the Mediterranean Sea: Fourier et al., 2020 and Pietropolli et al., 2023;

- use nitrate since it is the only variable common to all published architectures;

- use a common dataset (i.e., the subset of the test dataset) to perform a fair comparison among the different architectures.

The new results will be shown in a new Appendix and commented in the discussion section. Please refer to the point Rev#2.21 for the new Appendix B and modifications to the main text.

Finally, we will revise line 335 as follows:

"However, while MLP architectures **have demonstrated to**  provide good training and test errors **for point-wise input/output**(Pietropolli et al. (2023a); Fourrier et al. (2020); Bittig et al. (2018); Sauz.de et al. (2017)), **they can**  exhibit higher errors when predicting BGC-Argo profiles (Pietropolli et al. (2023a)**, Appendix B**)."

Rev#2.21 - l. 384 and l. 355-358 (btw. CANYON-MED with MLP architecture states a nitrate RMSE of 0.78 mmol m-3) will need correction, too.

We appreciate the reviewer's attention to the reported RMSE values for the CANYON-MED MLP and our comparison with the MLP model by Pietropolli et al. 2023. It is indeed correct that the CANYON-MED MLP (Fourier et al., 2020) states an RMSE of 0.78 mmol m-3. However, it's important to note that this RMSE metric is computed on a test set derived from the same dataset used for training the architecture, consisting of samples collected by ship cruise measurements. This context is distinct from the test set used in the present study, which comprises float measurements from the Argo-float dataset.

To ensure clarity and avoid any potential confusion, we will revise the relevant sections of our manuscript (lines 384 and 355-358) to explicitly state the context and nature of the datasets used for computing the RMSE values for PPCon and the two MLPs architectures (Fourier et al., 2020; Pietropolli et al., 2023). This clarification will provide an accurate comparison of the performance metrics across different models,in line with the reviewer's suggestion.

In particular, new Appendix B (Comparison between reconstructed nitrate profiles by PPCon and MLP architectures) will report examples of reconstructed profiles by the different approaches and the RMSE metric for PPCon and previous MLP applications. The RMSE is computed on the same dataset: the sub-set of profiles constituting the test dataset presented in section 2. This type of comparison is done only for nitrate that is the only variable computed by all three ML architectures.

The new Appendix B will be as follows:

**"Appendix B: Comparison between reconstructed nitrate profiles by PPCon and MLP architectures**

The present appendix aims to show the performance of three different ML architectures to reconstruct nitrate profiles that use Argo profiles of temperature, salinity and BGC-Argo profiles of oxygen. The three ML architectures are: the 1D CNN of the present work (PPCon), MLP trained on point-wise data from Emodnet (MLP, Pietropolli et al., 2023) and MLP trained on point-wise data (CANYON-Med, Fourier et al., 2020). The comparison is done on the sub-set of profiles used in the test phase described in section 2. Figure B1 shows some measured and reconstructed float profiles. The visual comparison reveals the higher performance of PPCon to match the shape of the measured profiles (e.g., depth and intensity of the nitracline) and the nitrate values of the deepest part of the profiles observed in the different Mediterranean sub-regions

The quantitative assessment of the performance of the three ML architectures is shown in Table B1 that reports the RMSE computed over all profiles of the sub-set used in the test phase. RMSE of the reconstructed profile by PPCon is more than 30% lower than that computed on the MLP reconstructions.

[Figure]

Fig. B1: Nitrate profiles from BGC-Argo dataset (green, measured) and reconstructed by PPCon (cyan dashed line), MLP as in Pietropolli et al. (2023) (purple dashed line) and CANYON-Med (dark blue dashed line). Profiles are selected from the sub-set used in the test phase of the present work. Float positions are as follows: 6901032 in NWM, 6903249 and 6903153 in ION, 69032904 in LEV and 6901767 in TYR and 6901769 in SWM.

|  | PPCon | CANYON-Med (Fourier et al., 2020) | MLP (Pietropolli et al., 2023) |
|---|---|---|---|
| Nitrate RMSE [mmol/m3] | 0.52 | 0.79 | 0.98 |

Table B1: RMSE of the three ML architectures computed over the nitrate profiles of the sub-set BGC-Argo dataset of the test phase."

Additionally, as reported above, old line 350-358 will be changed as follows:

"Although both MLPs and PPCon employ similar input information (date, geolocation, temperature, oxygen, and salinity), their treatment of this data differs significantly. **While the**

**current MLP applications** process the input and output **as point-wise data** , PPCon utilizes vector representations of the vertical profiles. This approach  effectively exploit**s** the potential of a 1D CNN, which intrinsically preserves the characteristic profile shape of the input and output variables Kiranyaz et al. (2021). When comparing the predictive performance of these techniques in generating vertical profiles from float data, distinct differences emerge. MLPs **can**  produce profiles **affected by**  artificial discontinuity , while the profiles generated by PPCon exhibit a smoother and more realistic appearance (**Appendix B2**). **Additionally**,  **the RMSE values computed on the reconstructed nitrate profiles of the test sub-set confirms the better performance of the 1D CNN approach with respect to a MLP approach trained on point-wise data** (**Appendix B2**).

Finally, old line 384 in conclusion will be changed as follow:

"The introduced model, named PPCon, utilizes a spatial-aware 1D CNN architecture that effectively learns the characteristic shape of the vertical profile, enabling precise and smooth reconstructions. PPCon represents a **potential**  advancement **in predicting BGC-Argo profiles** over previous  **on MLP applications,** which operate on point-wise  input and output"

I suggest to take out the MLP-irregularities-and-jumps claims entirely, as they are not sufficiently supported, and stick to the fact that CNNs can take benefit from data in their vicinity/neighorhood in a better way and explicitly during training (thanks to the conv/deconv layers).

Considering the reviewer' suggestion, we have decided to move the comparison between the three different architectures (i.e., PPCon, Canyon-Med and MLP by Pietropolli et al. 2023) from the main text to a new Appendix B (reported in the response to the comment above). Thus, the old figures 2 (new figure 3) of the main text will be re-drawn reporting only PPcon and measured BGC-Argo profiles and the result section will focus on the capability of PPCon to produce smooth and accurate profiles for the nitrate. A new Appendix B will report a more detailed comparison between the different architectures. Paragraphs in the discussion section will be revised accordingly. All these changes at the main text are detailed in the previous point.

- l. 87: "Array for Real-time Geostrophic Oceanography" This is an interesting fit to match "Argo", but Argo is not an abbreviation. (It is inspired by Greek mythology: https://argo.ucsd.edu/about/)

We thank the reviewer for pointing out the incorrect interpretation of the term *argo*. The old line 87 will be changed as follows, also considering one of the previous points.

"The data used to train and test the architecture discussed in this paper comes from the  **BGC-Argo** program (Bittig et al. (2019)), specifically the Argo float collecting also biogeochemical variables."

- The architecture/design of PPCon and their CNN-approach is hard to understand. The authors refer/cross-refence to different elements of their approach, without clear, concise wording. E.g., there are several references to the four point-wise inputs, the seven-channel tensor, or three variables (e.g., l. 150, 145, 143, 139, 131f.) without a clear sentence like: "Per profile, we have 4 point-wise inputs, which are latitude, longitude, (decimal??) year, and year day. In addition, we have three 1x200 input vectors for temperature, salinity, and oxygen profiles, respectively."

We understand the importance of presenting a straightforward explanation of how the various elements of our architecture integrate. To address this, we will insert a summarizing sentence immediately after line 143 in our revised manuscript to provide a clear overview of the architecture. The sentence will be as follows:

"The input to the PPCon architecture consists of four point-wise inputs — latitude, longitude, day, and year — which are transformed into a vectorial input using an MLP architecture. In addition, the architecture uses for the training three 1x200 input vectors representing the profiles of temperature, salinity, and oxygen."

- Can Figure 1 be modified so that it mirrors the informations from Table 1 and Table 2 on the specific PPCon architecture (e.g., layer sizes on the MLP; actual series of conv/deconv on the CNN)?

We appreciate the reviewer's suggestion, however the present version is a compromise between clarity and level of information. We would like to keep it simple by providing an illustration of the architectural components of the PPCon. We decided to slightly modify the figure by including more layers in the convolutional architecture and removing the "vectorial 1D CNN input". For sake of clarity, the figure has the aim of exemplifying the alternating pattern of convolutional and deconvolutional layers through the changing layer dimensions without representing the exact number of layers to avoid overcomplicating the visual representation. Similarly, the representation of the MLP in the figure was designed to exemplify the increasing number of neurons per layer, rather than to depict the exact layer sizes, which are substantial and would have complicated the figure's clarity.

Additionally, we will modify the caption to better describe the illustration and its components.

The new figure 1 will be follows:

[Figure]

The caption of new figure 1 will be as follows:

"Figure 1: Illustration of the principal architectural components of the PPCon model: i) **MLP network to transform the point-wise inputs (day, year, latitude, and longitude) into vectorial form; ii) vectorial inputs (profiles of temperature, salinity, and oxygen and output of the MLP); iii) structure of encoder-decoder of a 1D CNN architecture; iv) output vector representing the vertical profile of one of the target variables (nitrate, chlorophyll, or backscattering)."**

- l.232f: Why did the authors decide to not use an input normalization, which is a common approach, with the same advantages as mentioned for batch normalization? It would probably make the hyperparameters in Table 3 more similar to each other, too.

Our choice was informed by a series of preliminary experiments where we compared the performance of the PPCon architecture with both normalized and non-normalized inputs. Interestingly, we observed that normalizing the inputs did not yield any significant improvement in the model's performance. Based on these findings, we opted not to implement input normalization in the current iteration of our model.

The reviewer's point about the potential for normalization to harmonize hyperparameters across different models (nitrate, chlorophyll, and bbp700) is something we want to investigate. This aspect was not a primary consideration in our initial decision-making process. We acknowledge that normalization could contribute to more consistent hyperparameter settings across these models, potentially simplifying the architecture's configuration and enhancing its adaptability.

One possible advantage of not using normalization —when there is no decrease in performance, as in this case— is that it requires one fewer step for the integration into a pipeline.

- Why did the authors chose to use a separate MLP for each of the 4 point-wise inputs, to transform it from 1x1 to 1x200 shape? A 200x replication so that, e.g., latitude becomes a 1x200 sized vector with (constant) latitude per profile would have sufficed to concatenate it together with the 3 input profile vectors, with the 1D CNN alone then tasked to find an optimal representation/fitting.
If I interpret Table 1 correctly (1 input, 80 neurons in 1st hidden layer, 140 in 2nd, 200 in 3rd, 200 in output layer), there are ca. 80.000 parameters for each of the 4 MLPs alone. Again, I struggle to understand the actual size of the dataset used for training, but with in total approx. 120.000/70.000 chla/bbp or nitrate BGC-Argo profiles worldwide, and given that we consider a training dataset within the Med Sea, I estimate the number of profiles to be somewhere around 3.000-5.000 or smaller. The MLPs seem to me like a very badly constrained task, even for machine learning and a decent dropout rate. It's an awfully complex MLP just to get sth. from 1x1 to 1x200 shape, which is then fed into yet another neural network.

While it is true that the number of parameters of the MLP used to convert from scalar to 1D inputs are high, we did not consider this fact a concern mainly for two reasons:

1) Overparameterization is one aspect where it appears that neural networks are able to learn in a way that generalizes well even when the number of parameters is higher than the number of training samples. While the reasons for this effect are still not completely understood from a theoretical point of view, there are multiple works in the area, e.g., Arora et Al. (2018).

2) Most importantly, the MLP component of the network only has access to the scalar data in the forward pass, which are insufficient by themselves for the reconstruction of an entire profile or even to distinguish between different profiles. Hence any amount of memorization of the training data in the weight of the MLPs would have limited impact, which is confirmed by the fact that the architecture generalizes well when employed on test data.

In more details, the main motivation behind the use of a MLP for transforming scalar inputs into vectorial ones is that the same scalar value can have different impacts at different depths and this can be learned by the MLP performing the 1x1 to 1x200 transformation (we justify this choice in line 137 of the manuscript). In particular, while not widespread, a similar approach was already applied in "Learning Hand-Eye Coordination for Robotic Grasping with Deep Learning and Large-Scale Data Collection" by Levine et Al. (2016). In that paper the authors used a fully connected layer with 64 units whose outputs were added pointwise to 64 response maps (i.e., they provided the equivalent of a bias term). In our case we still employ fully connected layers but, instead of performing a pointwise addition to multiple response maps, we directly add the output as a channel. In both cases the idea of using a MLP to adapt the data before inserting it in the CNN architecture is present.

Reference used for the response:

- Levine, S., Pastor, P., Krizhevsky, A., Ibarz, J., & Quillen, D. (2018). Learning hand-eye coordination for robotic grasping with deep learning and large-scale data collection. *The International journal of robotics research*, 37(4-5), 421-436.

- Arora, S., Ge, R., Neyshabur, B., & Zhang, Y. (2018, July). Stronger generalization bounds for deep nets via a compression approach. In *International Conference on Machine Learning* (pp. 254-263). PMLR.

- On a similar note: Can the authors please provide information on the amount of parameters (i.e., how flexible the entire PPCon is) vs. the number of profiles for training (i.e., data constraints) so that a reader gets a better idea of how well constrained PPCon is in general?

The number of trainable parameters in our PPCon model consists of 39,700 for each scalar input and 253,249 for the convolutional part, totaling 412,049 parameters. This parameter structure ensures comprehensive learning capability while maintaining model efficiency. For training, we used 80% of the profiles from our dataset, as detailed in Table 3 of the manuscript. This amounts to 2,337 profiles for nitrate, 3,189 for chlorophyll, and 3,952 for bbp700.

- Did the authors try to exclude the year from the inputs, with what effect? Given the training data covers only 6 years, it would be very surprising to me if the "year" input had a lot of explanatory power.

The choice to include the year was a legacy from the MLP approaches, where the year is an input. Even if the used BGC-Argo dataset covers only 6 years, the PPCon is thought to be applied with a longer dataset upon available. We have not tested this option, however as for other aspects (for example the one mentioned in the previous point), the public release of the PPCon software will allow to test sensitivity of the results to architecture adjustments.

We will add a sentence in the "Code and data availability" section

**"In the present work, we present an optimized version of the architecture for the specific dataset of the Mediterranean Sea, but the release of the PPCon code allows arbitrary adjustments of the architecture."**

- Eq. 2: Is the hyperparameter alpha missing from the equation?

We appreciate the reviewer's observation regarding the omission of the coefficient in Equation (2). To rectify this, we have now included the missing coefficient in Equation (2). Additionally, to avoid any confusion with the symbol used in Equation (1), we have chosen a different symbol for the coefficient both in the equation and the main text.

- There needs to be an evaluation against existing methods! Several ones are quoted in the well-written introduction, but they do not appear later in the manuscript. In particular, CANYON-MED, as being of similar scope and specifically trained on the Med Sea, too, is a prime candidate for comparison/evaluation (at least for nitrate). The authors evaluate PPCon against an MLP, the work of Pietropolli et al. 2023a, briefly mentioned, which is by the same authors? (1) This must be made clear on the figures/text and (2) at least CANYON-MED predicted nitrate profiles need to be added in the comparison, both on the individual

examples as well as for the overall validation RMSE. (If there exist chla/bbp predictions suitable for the Med Sea, too, they should be added - but I am presently not aware of any.)

As reported in one of the previous points, we decided to include the new appendix B dedicated to the comparison of the PPCon with previous reconstructing methods (namely CANYON-MED, Fourier et al., 2020; and MLP, Pietropolli et al., 2023) optimized for the Mediterranean Sea. The comparison is shown only for nitrate, as there is no output for bbp700 and chlorophyll by both the other two methods for the Mediterranean Sea.

- l. 358: CANYON-MED states a nitrate RMSE of 0.78 mmol m-3.

Thanks for pointing out this aspect. The nitrate quality (i.e., RMSE) of MLP architectures based on point-wise input and output data is 0.78 mmol m-3 (Fourier et al., 2020) and 0.50 mmol m-3 (Pietropolli et al., 2023).

On the other hand, when MLP architectures are used to reconstruct nitrate profiles the RMSE can be different.

In fact, as explained above, we conducted a robust comparison among the different methods by using the same set of reconstructed profiles (i.e., the sub-set of test profile of the BGC-Argo dataset). Results are shown in the new Appendix B (see previous Rev#2.21 point) and the RMSEs are 0.79 mmol/m3 and 0.98 mmol/m3 for CANYON-MED and MLP, respectively.

Together with the new Appendix B we will change the sentence at old line 358 as follows:

"This improvement is confirmed also by **the RMSE values computed on the reconstructed nitrate profiles of the test sub-set confirms the better performance of the 1D CNN approach with respect to a MLP approach trained on point-wise data (Appendix B2).** , which is lower when using the PPCon model (RMSEPPCon = 0.61) compared to the state of the art of MLP architectures RMSEMLP = 0.87 according to Pietropolli et al. (2023a))"

- Please add the float cycle number (i.e., identification of which profile of a given float deployment) to the example profiles (Table 5 and Figure panel titles) so that it becomes clear (and easier to redo/recalculate) for a reader which profile was used.

We thank the reviewer for the suggestion, we will add the information related to the cycle number to all the profiles displayed in (old number) Figures 2, 3 and 4.

[Figure]

- Figure 3: All of the Chla examples are of a deep Chla maximum (DCM) shape. At least one example should be a winter deep mixing example, which occurs in the Med Sea, for completeness. Otherwise one could argue that Chla should be at least as 'easy' to predict as

nitrate, because the shape is always of a DCM-kind (-> l. 287: If Chla had always a DCM profile shape, then... ).

We thank the reviewer for highlighting the omission of a deep mixing example in our discussion of the results. To address this and provide a more complete list of examples of chlorophyll profiles, we will revise Figure 3 including one case of chlorophyll winter bloom instead of one of the existing profile examples.

The new figure 4 (old figure 3) will be as follows:

[Figure]

**New Figure 4 (old Figure 3)**: Profiles of chlorophyll for some selected floats (WMO numbers **and cycle** in the title) and dates. **Profile dates and geolocations are reported in Table 5.** Comparison between measured profile (green lines) and PPCon reconstruction (blue dashed lines). Profiles are from the subset used for the test.

- l. 286f: "Higher quality in the prediction is achieved for nitrate, followed by chlorophyll and bbp700" - How was this judged/obtained?

This was a qualitative consideration after the visual inspection of all the reconstructed profiles of the test dataset. By looking at the generated profiles we noticed that the one predicted with higher similarity (e.g., depth and steepness of nitracline, the values in the deeper layers, depth and intensity of the DCM) were the ones produced by nitrate, as most of the times is more similar to the original. In order to be more precise we will modify the sentence as follows:

**"the visual inspection of all test profiles (not shown) revealed that the** higher quality in the prediction is achieved for nitrate, followed by chlorophyll and bbp700"

- l. 371: [...] application on the GDAC's *BGC*-Argo *Med Sea* float dataset [...]

The sentence will be changed as follows:

**"For instance, the application of PPCon on Argo and oxygen profiles in the Mediterranean Sea for the period**  from 2013 to 2020 enabled the generation of **5234 (nitrate), 3879 (chlorophyll), 3307 (bbp700)**

synthetic  profiles,  which means doubling the chlorophyll and bbp700 BGC-Argo profiles and more than tripling those of nitrate."

- l. 365: "cloud coverage" and "incomplete swaths"??

According to the reviewer suggestion we will modify the sentence in lines 364-366 as follows:

"Surface satellite observations are limited by cloud  coverage and incomplete  swaths of satellite sensors (Donlon et al. (2012)), while profiling the ocean interior is limited by the capacity of deploying and retrieving sensors and measurements with sufficient coverage."

**NEW TABLES AND FIGURES**

| | WIN TRAIN | WIN TEST | SPR TRAIN | SPR TEST | SUM TRAIN | SUM TEST | AUT TRAIN | AUT TEST |
|---|---|---|---|---|---|---|---|---|
| NITRATE | 0.51 | 0.51 | 0.51 | 0.52 | 0.51 | 0.49 | 0.48 | 0.51 |
| CHLA | 0.08 | 0.07 | 0.12 | 0.13 | 0.08 | 0.08 | 0.05 | 0.05 |
| BBP700 (x10^-4) | 2.6 | 2.4 | 2.3 | 2.6 | 1.5 | 1.4 | 1.5 | 1.5 |

Table 6. RMSE calculated between the float measurements and the reconstructed values obtained from the PPCon architecture for all variables inferred. This metric is evaluated individually for the train and test sets. The RMSE is computed for different seasons of the year (described in Section 2).

| | NWM TRAIN | NWM TEST | SWM TRAIN | SWM TEST | TY TRAIN | TY TEST | IO TRAIN | IO TEST | LEV TRAIN | LEV TEST |
|---|---|---|---|---|---|---|---|---|---|---|
| NITRATE | 0.62 | 0.65 | 0.37 | 0.38 | 0.44 | 0.44 | 0.41 | 0.41 | 0.48 | 0.51 |
| CHLA | 0.14 | 0.13 | 0.10 | 0.12 | 0.08 | 0.08 | 0.04 | 0.04 | 0.05 | 0.05 |
| BBP700 (x10^-4) | 2.6 | 2.4 | 2.1 | 2.0 | 2.3 | 2.3 | 1.4 | 1.6 | 1.4 | 1.7 |

Table 7. RMSE calculated between the float measurements and the reconstructed values obtained from the PPCon architecture for all variables inferred. This metric is evaluated individually for the train and test sets. The RMSE is computed for different geographic areas of the year (described in Section 2).

| nitrate | nitrate | chla | chla | bbp700 | bbp700 |
|---|---|---|---|---|---|
| 6901648 | 0.70 | 6901648 | 0.14 | 6901649 | 1.9 x 10^-4 |
| 6901764 | 0.31 | 6901496 | 0.13 | 6901496 | 2.2 x 10^-4 |

Table 8. RMSE calculated between the float measurements and the reconstructed values obtained from the PPCon architecture over the external validation dataset.

[Figure]

Figure 6. Hovmöller diagrams for the nitrate of two selected floats (WMO name in the title) belonging to the external validation set. BGC- Argo measurements (upper panels) and PPCon prediction (lower panels) are compared. WMO 6901648 sampled the 40°N − 42°N and 2°E − 6°E area during 2014 − 2016, whereas WMO 691764 sampled the 31°N − 34°N and 26°E − 40°E area during 2015 − 2017.

[Figure]

Figure 7. Hovmöller diagrams for the chlorophyll of two selected floats (WMO name in the title) belonging to the external validation set. BGC-Argo measurements (upper panels) and PPCon prediction (lower panels) are compared. WMO 6901648 sampled the 40°N − 42°N and 2°E − 6°E area during 2014 − 2016, whereas WMO 6901496 sampled the 42°N − 43°N and 7°E − 12°E area during 2013 − 2014.

[Figure]

Figure 8. Hovmöller diagrams for the bbp700 of two selected floats (WMO name in the title) belonging to the external validation set. BGC- Argo measurements (upper panels) and PPCon prediction (lower panels) are

compared. WMO 6901649 sampled the 39°N − 41°N and 3°E − 7°E area during 2014 − 2016, whereas WMO  **6901496 sampled the 42°N − 43°N and 7°E − 12°E area during 2013 − 2014**.

---

## Referee Report (RR1)

Dear Gloria Pietropolli, Luca Manzoni, and Gianpiero Cossarini,

I would like to thank the authors for their efforts to improve the manuscript. It now reads much more smoothly and the consistent use of terminology to describe items/aspects make it much easier understandable.

The Med Sea focus is still a bit hidden, only to appear in the last sentence of the abstract. I would prefer this info to be presented more up-front, but I guess this is at the author's discretion.

However, there is one aspect where I dissent with what the authors claim. They write:

> l. 8f. [1]: "However, MLPs lack awareness of the typical shape of biogeochemical variable profiles they aim to infer" ('claim *a*')

and continue:

> l. 9f.: "resulting in irregularities such as jumps and gaps when used for the prediction of vertical profiles.". ('claim *b*')

As written in my previous comment, (1.) I don't think either of the two statements is true, and (2.) I don't see evidence presented by the authors to convince me otherwise (more on 2. later).

I see and appreciate in the remainder of the manuscript that these two claims have been toned down, compared to the initial version. However, I don't think they are justifiable.

To **claim *a***: The authors are correct in that MLPs act and are trained on point-wise data, whereas CNNs take strides of data (e.g., profiles) and consider both their value and their arrangement (e.g., profile shape). To use CNNs to predict some profile data from other profile data, taking benefit of the shape of that other profile data, is a (promising) step forward compared to MLPs predicting some profile data.

However, that does not mean that MLPs for Ocean prediction are agnostic to their neighbouring data points. Instead of an explicit shape awareness like CNNs, MLPs have an implicit awareness of the typical shape of profiles they want to infer. Why?

Because the Ocean is smooth.

The parameter space in the Ocean is continuous and smooth (with a large thanks to mixing). Just for illustration: Below two T-S diagrams for two of the floats used. (Quality-controlled) Ocean data are smooth to start with.

[Figure]

Going along a profile step-by-step (both in parameter space or against depth) gives only small, step-wise modifications of the variable to be predicted and the variables used as predictors alike. Which means that, at any given point in parameter space, you have a certain idea of how your environment will look like: Probably not an awful lot different, but just a little. I.e., even if only given point-wise knowledge at a time, MLPs do have an awareness of how their profile will look like nearby (i.e., not a
* * *
[1] Line numbers refer to the tracked-changes manuscript version "egusphere-2023-1876-ATC2.pdf".

lot different). I.e., there is (some) spatial awareness of profile shape in MLPs, too, due to point-wise proximity in parameter space and due to the Ocean's parameter space being smooth.

To **claim *b***: Starting off from (*i*) a smooth parameter space (previous point), and (*ii*) using a neural network (MLP) architecture complexity suited for the size of the training data as well as (*iii*) ensuring that there is no overfitting by properly selected regularization, then the (MLP) neural network outcome must be an approximation of the parameter space trained on. If properly regularized, the neural network is by definition smoother than the training data from (*i*). If not overly complex, then the network's weights and parameters are sufficiently constrained by the training data from (*i*), so that the (MLP) network represents a continuous function (no poles, irregularities, gaps). If the training data are smooth to start with, then that cannot cause irregularities or gaps either.

So I am left with claims where my arguments run against them. The two claims are brought up again in l. 68-73 but without illustration. So when coming to Figure 3 (nitrate profiles for selected floats), I wondered why the measurements and PPCon are shown but the Pietropolli et al. 2023a MLP comparison (as well as other MLPs for comparison like CANYON-MED, CANYON-B) were dropped?

Thank you for adding the WMO and profile numbers, which allowed me to go search for the data and do those comparison plots on my side. Here's what I get for the measurements as well as the two MLPs where I had access to the code for prediction: CANYON-MED (specifically trained on the Med Sea with a dataset extended beyond GLODAPv2) and CANYON-B (trained with very little Med Sea data from GLODAPv2, so no great performance to expect; but it provides confidence intervals, which I find instructive):

[Figure]

Same as Fig. 3 but with measured float data directly from the GDAC and two MLPs added (dark blue: CANYON-MED, light blue: CANYON-B).

From the comparison, I find it rather comforting and confirming my line of argument that the MLPs give a similarly smooth profile shape as the CNN (in the manuscript Fig. 3) and no irregularities or jumps in the profile – unless governed by the water mass properties (e.g., middle profile just shallower than 50 dbar) and seen in the measured nitrate, too. (As said, I could not confirm/falsify whether the EMODNET-based Pietropolli et al. 2023a MLP shows irregularities or jumps in the profile.)

What I find a bit discomforting is that the measured data in my case looks not the same as shown in Figure 3 of the manuscript: All profiles in Fig. 3 are much smoother (e.g., base of the mixed layer is significantly eroded; interleaved water mass around ~50 dbar in middle panel fully absent) and also the

bottom portion of the middle profile has a different shape (and value). I have confirmed the profile's locations and dates that I used – they match to the floats and cycle numbers (Thank you for Table 5)!

The two claims are then taken up again in the results (l. 317-318; no illustration) and later on in the discussion (l. 411-412; no illustration) and the reader is referred to Appendix B (l. 392; l. 414).

This Appendix B (comparison between reconstructions by PPCon and MLP architectures) is very promising. And after experience with Fig. 3 I tried to redo the figure B1 on my end, too:

[Figure]

Same as Fig. B1 but with measured float data directly from the GDAC and two MLPs added (dark blue: CANYON-MED, light blue: CANYON-B).

From my perspective, both MLPs do an excellent job in reproducing the measured profile (even CANYON-B, for which no great things should be expected in the Med Sea). Both MLPs even include proper reproduction of variability caused by interleaved water masses along the vertical profile.

However, I see a couple of discrepancies to the manuscript Fig. B1:

- The measured data is again of different shape, value, and smoothness – for all profiles shown. None of the fine scale features are visible in the manuscript (from what I can eye-ball).
- The fine scale features are pretty much mirrored in CANYON-MED; and CANYON-MED seems to match between my figure and manuscript Fig. B1, unlike the measured data. The CANYON-MED MLP is also spot-on on most of the measured profile (in my figure; not in the manuscript Fig. B1).
- The float 6903153 in the ION (lower left panel) does not carry a nitrate sensor at all.

These aspects are quite discomforting and raise an eyebrow on whether there are similar discrepancies in other parts of the data, and on which data were compared with what measurements with respect to performance (RMSE) of the various ANNs, both PPCon and others.

There are a few instances throughout the manuscript, where WMO numbers got a bit scrambled, so it might be as simple as incorrect WMOs and cycles labelled. Or, there may be a more profound issue with

the data used, or a mess-up in measured profiles compared to actually different MLP/PPCon predicted profiles. This would be dramatic. In any case, these issues need to be addressed before a publication.

(a) Where do the discrepancies in the measured data come from? How large is the extent of the discrepancies? What's the impact on the method and its evaluation?

(b) Both from the line of argument presented in this comment, as well as from the illustrations (reproduced Figs. 3 and B1), I cannot recognize that either of claim *a* or *b* can stand. (Rather, MLPs can do a surprisingly good job in reproducing profile shape thanks to (point-wise) water mass characteristics.)

I would therefore ask you to remove those claims entirely, or to substantiate them. (Again, my perspective neglects the Pietropolli et al. 2023a MLP, where I don't have access to the code. It may be that some of your claims *a* or *b* may apply to and be true for the Pietropolli et al. 2023a MLP. But then it cannot take hostage of all MLPs, given that it doesn't apply to other MLP models or architectures.)

(c) Could it be that fine scale characteristics of the vertical profile, like interleaved water masses, cannot be well reproduced by the CNN-based PPCon model/architecture? Because it puts more emphasis on the entire/large-scale profile shape and 'neglects' the information from the water mass properties, which is the only information available to MLPs?

(d) Why are the measured data in the manuscript without fine scale and with mostly eroded mixed layer base?

**Further points:**

- Figure 1: I cannot find float 6901767, the second nitrate float used later on (Fig. 7 and Tab. 8). Or is it a WMO mix-up there (and it's 6901648??).

- l. 133: "Table 2" referred to is missing.

- l. 134: From the description of Amadio et al. 2023 (and the above experience) it reads that the float data were treated with the "bit.sea python package (Bolzon et al., 2023)", including "a smoothing task". For a manuscript that focuses and puts emphasis on profile shapes, this is important information that belongs into the description of the dataset (and cannot be hidden somewhere inside some reference). And which warrants some (brief) discussion, because it seems that your figure's mixed layer bases are much more eroded than in the original float data. Which may have implications if PPCon were to be used for augmenting a float dataset.

- section 3.1: The question remains on why such a complex architecture was chosen and what should be gained from 4 separate MLPs per singular input (over replication of input data into a 200x1 vector each) to be fed into the CNN. But I won't insist and just note that this is still weakly motivated.

- l. 182: "adding zero padding to the borders of the input tensor": This requires the input to be normalized, i.e., mean data subtracted (and ideally divided by the standard deviation). A batch normalization is mentioned only for the output tensors in the text, not for the input tensor. From table 2 I understand that batch normalization (BN) is done for every layer's input, is it? So maybe follow the logic as it is outlined in the table also in this paragraph: Layers consist of BN, SELU and have Dropout. To compress/decompress information we have kernels/strides, which require padding with ...

- l. 314: "For the nitrate variable, the reconstruction performed by the MLP model is also reported (Pietropolli et al., 2023a)." No, it's not – but it should! :-)

For Figure 3 and also Figure B1, I'd also suggest to use a bit wider spectrum of colour than different shades of blue, to be able to better distinguish the different models.

- l. 317f: "...than the previous MLP architecture by Pietropolli et al. (2023a), but similar to MLPs by Fourrier et al. or Bittig et al."? But my general recommendation would be to drop the second part of the sentence.

- l. 472: That predictions tend to better in deep waters compared to surface waters is true for of all approaches; it's not unique to PPCon. It's a feature due to the different variability of the Ocean's parameter space at depth vs. at the surface, nothing of a particular model's architecture.

- Fig. B1 caption: 6903153 in ION has no nitrate sensor.

Minor points:

- l. 42: replace "frequency" by something more fitting? Maybe

- l. 133: "Table 2" referred to is missing.

- l. 145: sample date?

- l. 303: "in the ION[+, SWM, and TYR,] with RMSE values below 0.5 [+unit]"?

- l. 306f.: check "which are the highest ..." – Does this still apply?

- l. 395: "DT" was this abbreviation introduced? Why not spell it out

Typo's:

- quite a few instances: "BCG" instead of "BGC"

- l. 41: closing ")"

- l. 263: $\alpha_s$ ?

- l. 377: closing ")"

- l. 403: understand

- l. 439; BGC-Argo network

- Fig. B1 caption: 6902904

- l. 102/l. 509: Cross-check the Bittig et al. 2019 reference. That's the one you intended? (technical documentation vs. a published manuscript?)

---

## Author Response (AR2)

**PPCon 1.0: Biogeochemical Argo Profile Prediction with 1D Convolutional Networks**

**Response to the Editor**

Dear Editor and Reviewers,
We appreciate the constructive comments and suggestions provided by the reviewer. Following their suggestions, we present our point-by-point responses to the reviewer's comments.
The reviewer's comments are highlighted in blue, followed by our responses in black. In each response, we detail the proposed changes to the manuscript, including any modified text and/or figures (in red).

Dear Gloria Pietropolli, Luca Manzoni, and Gianpiero Cossarini,

I would like to thank the authors for their efforts to improve the manuscript. It now reads much more smoothly and the consistent use of terminology to describe items/aspects makes it much easier to understand.

The Med Sea focus is still a bit hidden, only to appear in the last sentence of the abstract. I would prefer this info to be presented more up-front, but I guess this is at the author's discretion.

**(R1)** However, there is one aspect where I dissent with what the authors claim. They write: l. 8f.: "However, MLPs lack awareness of the typical shape of biogeochemical variable profiles they aim to infer" ('claim a') and continue: l. 9f.: "resulting in irregularities such as jumps and gaps when used for the prediction of vertical profiles." ('claim b')

(**) We thank the reviewer for these observations, we will discuss them further and make the appropriate changes in the text.

Nevertheless, before proceeding, we would like to point out that the goal and key messages of our paper target three variables: nitrate, chlorophyll and bbp700 The real strength of convolutional architectures (i.e., explicitly considering 1D vectors) compared to MLP architectures becomes evident at least for the last two, where convolutional architectures can learn the shape of the vertical profile, making it possible to predict more vertically-dynamic variables like chlorophyll and bbp700..

Regarding the nitrate variable, based on our experience with training and testing MLPs and CNNs, we noted that CNNs can be a viable alternative to previously proposed MLP approaches. These considerations include both the performance obtained with CNNs and the less demanding process of creating the architecture. Creating an MLP architecture requires more oceanographic domain knowledge (e.g., the preprocessing in CANYON-MED, where each input needs specific functions) and a longer hyperparameter tuning phase, while CNNs are naturally suited for this kind of application, making the tuning phase less demanding.

Said that, we agree with the reviewer that MLP approaches for nitrate have shown very good performance, so the effectiveness of CNN and MLP approaches is similar.

Regarding the two claims:

**Claim a:** MLPs, from a theoretical point of view, operate point-wise. However, we agree that if the input changes smoothly, MLPs will provide a smooth output. On the other hand, 1D-CNNs, by construction, can deal with profiles. Our objective is to demonstrate that 1D-CNNs can work as well as previously proposed approaches (again, when referring to the nitrate variables). We will rephrase the sentence in the abstract as follows: "Although MLPs can produce smooth outputs if the inputs change smoothly, 1D-CNNs are inherently designed to handle profile data effectively."

**Claim b:** The sentence applies not only to nitrate but to all variables from BGC-Argo. Chlorophyll and bbp700 can benefit significantly from a 1D-CNN approach, avoiding undesired jumps in profile reconstruction (see Figures 4 and 5 and figures in Pietropolli et al., 2023a). However, we agree with the reviewer that for nitrate, MLPs perform as smoothly as 1D-CNNs. We will change the sentence in the abstract (lines 8-10 of the previous version) as reported in the previous point.

As written in my previous comment, (1.) I don't think either of the two statements is true, and (2.) I don't see evidence presented by the authors to convince me otherwise (more on 2. later).

I see and appreciate in the remainder of the manuscript that these two claims have been toned down, compared to the initial version. However, I don't think they are justifiable.

To claim a: The authors are correct in that MLPs act and are trained on point-wise data, whereas CNNs take strides of data (e.g., profiles) and consider both their value and their arrangement (e.g., profile shape). To use CNNs to predict some profile data from other profile data, taking benefit of the shape of that other profile data, is a (promising) step forward compared to MLPs predicting some profile data. However, that does not mean that MLPs for Ocean prediction are agnostic to their neighbouring data points. Instead of an explicit shape awareness like CNNs, MLPs have an implicit awareness of the typical shape of profiles they want to infer. Why? Because the Ocean is smooth.

The parameter space in the Ocean is continuous and smooth (with a large thanks to mixing). Just for illustration: Below two T-S diagrams for two of the floats used. (Quality-controlled) Ocean data are smooth to start with. Going along a profile step-by-step (both in parameter space or against depth) gives only small, step-wise modifications of the variable to be predicted and the variables used as predictors alike. Which means that, at any given point in parameter space, you have a certain idea of how your environment will look like: Probably not an awful lot different, but just a little. I.e., even if only given point-wise knowledge at a time, MLPs do have an awareness of how their profile will look like nearby (i.e., not a lot different). I.e., there is (some) spatial awareness of profile shape in MLPs, too, due to point-wise proximity in parameter space and due to the Ocean's parameter space being smooth.

To claim b: Starting off from (i) a smooth parameter space (previous point), and (ii) using a neural network (MLP) architecture complexity suited for the size of the training data as well as (iii) ensuring that there is no overfitting by properly selected regularization, then the (MLP) neural network outcome must be an approximation of the parameter space trained on. If properly regularized, the neural network is by definition smoother than the training data from (i). If not overly complex, then the network's weights and parameters are sufficiently constrained by the training data from (i), so that the (MLP) network represents a continuous function (no poles, irregularities, gaps). If the training data are smooth to start with, then that cannot cause irregularities or gaps either.

So I am left with claims where my arguments run against them. The two claims are brought up again in l. 68-73 but without illustration. So when coming to Figure 3 (nitrate profiles for selected floats), I wondered why the measurements and PPCon are shown but the Pietropolli et al. 2023a MLP comparison (as well as other MLPs for comparison like CANYON-MED, CANYON-B) were dropped?

We decided to present the comparison with other methods in Appendix B for several reasons. The most important reason is that this paper aims to predict three variables: nitrate, chlorophyll, and bbp700. The comparison with other methods is only possible for the nitrate variable, as there are no MLP architectures capable of adequately predicting the other two variables. Therefore, since nitrate is the only variable that allows for a meaningful comparison, we chose to include this information in the Appendix.

Thank you for adding the WMO and profile numbers, which allowed me to go search for the data and do those comparison plots on my side. Here's what I get for the measurements as well as the two MLPs where I had access to the code for prediction: CANYON-MED (specifically trained on the Med Sea with a dataset extended beyond GLODAPv2) and CANYON-B (trained with very little Med Sea data from GLODAPv2, so no great performance to expect; but it provides confidence intervals, which I find instructive):

Same as Fig. 3 but with measured float data directly from the GDAC and two MLPs added (dark blue: CANYON-MED, light blue: CANYON-B).

From the comparison, I find it rather comforting and confirming my line of argument that the MLPs give a similarly smooth profile shape as the CNN (in the manuscript Fig. 3) and no irregularities or jumps in the profile – unless governed by the water mass properties (e.g., middle profile just shallower than 50 dbar) and seen in the measured nitrate, too. (As said, I could not confirm/falsify whether the EMODNET-based Pietropolli et al. 2023a MLP shows irregularities or jumps in the profile.)

**(R2)** What I find a bit discomforting is that the measured data in my case looks not the same as shown in Figure 3 of the manuscript: All profiles in Fig. 3 are much smoother (e.g., base of the mixed layer is significantly eroded; interleaved water mass around ~50 dbar in middle panel fully absent) and also the bottom portion of the middle profile has a different shape (and value). I have confirmed the profile's locations and dates that I used – they match to the floats and cycle numbers (Thank you for Table 5)!

Thank you for pointing out that the bottom portion of the middle profile of Figure 3 has a different shape and value. After double-checking, we noticed a typo in the profile name (i.e.,

6901769 instead of 6901768). We have updated Figure 3 by changing the float name to the correct one (6901769_083).

(*) Regarding the second concern about the smoother profiles in Figure 3, this is due to the discretization performed on these vertical profiles. Specifically, for the nitrate variable, we considered profiles ranging from 0 to 1000 meters, and for the other two variables, we considered profiles ranging from 0 to 200 meters. We used equal discretization for the input and output which was necessary for the functionality of the convolutional architecture. Thus, a preliminary task of interpolation from the irregular and varying profile depths of original data to a regular discretization was performed.

We discretized the vertical profiles into 200 points, resulting in a 1-meter resolution for chlorophyll and bbp700 and a 5-meter resolution for nitrate. The 5-meter resolution, especially in the upper part, led to a smoother appearance compared to the original vertical profile, where the sampling was higher than once every 5 meters.

The discretization and interpolation are well explained in lines 222-225 in the "experimental setting" section and all reconstructed profiles (either MLP and CNN) in Appendix B have been produced with the same discretization to allow a fair comparison.

We will better clarify this point by adding a sentence in Appendix B (old lines 405-407) as follows:

The three ML architectures are: the 1D CNN of the present work (PPCon), MLP trained on point-wise data from Emodnet (Pietropolli et al., 2023a) and MLP trained on point-wise data (Fourrier et al., 2020). Input data from Argo and BGC-Argo for all approaches have been interpolated to the regular 5-meter discretization as explained in the section 4.2 Experimental Settings.

The two claims are then taken up again in the results (l. 317-318; no illustration) and later on in the discussion (l. 411-412; no illustration) and the reader is referred to Appendix B (l. 392; l. 414).

This Appendix B (comparison between reconstructions by PPCon and MLP architectures) is very promising. And after experience with Fig. 3 I tried to redo the figure B1 on my end, too:

Same as Fig. B1 but with measured float data directly from the GDAC and two MLPs added (dark blue: CANYON-MED, light blue: CANYON-B).

From my perspective, both MLPs do an excellent job in reproducing the measured profile (even CANYON-B, for which no great things should be expected in the Med Sea). Both MLPs even include proper reproduction of variability caused by interleaved water masses along the vertical profile.

However, I see a couple of discrepancies to the manuscript Fig. B1:

1. The measured data is again of different shape, value, and smoothness – for all profiles shown.

The explanation for this behavior is the same as reported in (R2). The smoothness of the measured profiles in Figure B1 differs slightly from the plots reported by the reviewer due to the discretization performed. Regarding the shape and value, the profiles reported in the paper match those in the reviewer's figure, except for the differences caused by the discretization.

2. None of the fine scale features are visible in the manuscript (from what I can eye-ball).

Again, this is due to the discretization performed, as detailed in (R2).

3. The fine scale features are pretty much mirrored in CANYON-MED; and CANYON-MED seems to match between my figure and manuscript Fig. B1, unlike the measured data. The CANYON-MED MLP is also spot-on on most of the measured profile (in my figure; not in the manuscript Fig. B1).

The measured profiles reported in the appendix and the figure provided by the reviewer match in shape and bottom values. The only difference is in the discretization, which results in smoother profiles in the appendix.

Regarding CANYON-MED, the difference arises because we performed the comparison using the first version of the software, which was available when we submitted this paper to the GMD journal. After considering the reviewer's comments, we noticed that the difference in the results is due to the different version of CANYON-MED employed (v1 in our case). Version 2 implemented in Python has recently been released, which provides more accurate predictions due to improvements in its implementation.

This highlights another strength of the convolutional architecture compared to the MLP architecture: the tuning of hyperparameters in MLPs is more time-consuming and significantly affects overall performance, while CNN can be (at least for some datasets) more resistant to some changes in the architecture and hyperparameters.

4. The float 6903153 in the ION (lower left panel) does not carry a nitrate sensor at all.

Thank you for pointing out that float 6903153 does not carry a nitrate sensor. After double-checking, we noticed a typo in the float name, we apologize for the mistake. We have updated Figure B1 by changing the float name to the correct one. The correct float cycle is 6901772_145.

These aspects are quite discomforting and raise an eyebrow on whether there are similar discrepancies in other parts of the data, and on which data were compared with what measurements with respect to performance (RMSE) of the various ANNs, both PPCon and others.

After correcting the two errors in reporting the float cycle numbers, we double-checked all other results in the paper. We confirm that the data are correct and accurately reported.

There are a few instances throughout the manuscript, where WMO numbers got a bit scrambled, so it might be as simple as incorrect WMOs and cycles labelled. Or, there may

be a more profound issue with the data used, or a mess-up in measured profiles compared to actually different MLP/PPCon predicted profiles. This would be dramatic.

As aforementioned, the problem was solely related to incorrectly reporting the WMO numbers in two cases. After correcting these errors, we have verified that all other data in the manuscript is accurate and correctly reported.

In any case, these issues need to be addressed before a publication.

(a) Where do the discrepancies in the measured data come from? How large is the extent of the discrepancies? What's the impact on the method and its evaluation?

As mentioned previously (R2), the smoother profiles in Figure 3 are due to the discretization performed on these vertical profiles. The vertical profiles of three variables were  discretized into 200 points, resulting in a 1-meter resolution for chlorophyll and bbp700 and a 5-meter resolution for nitrate. We acknowledge that this resolution of nitrate, especially in the upper part, led to a smoother appearance compared to the original vertical profile, where the sampling was higher than once every 5 meters.

The other discrepancy that the reviewer noticed where simply due to two WMO which were not reported correctly (middle profiles of Figure 3 and ION profiles of Figure B1). We update the correct WMO on the top of these profiles, and double-check all the others WMO.

Therefore, the extent of the discrepancies is limited to the smoothing effect introduced by averaging the vertical profiles to a consistent resolution. The impact on the method and its evaluation is minimal in terms of the overall performance metrics (RMSE) because the discretization is uniformly applied across all profiles, ensuring that the comparative evaluations remain consistent and fair.

(b) Both from the line of argument presented in this comment, as well as from the illustrations (reproduced Figs. 3 and B1), I cannot recognize that either of claim *a* or *b* can stand. (Rather, MLPs can do a surprisingly good job in reproducing profile shape thanks to (point-wise) water mass characteristics.)
I would therefore ask you to remove those claims entirely, or to substantiate them. (Again, my perspective neglects the Pietropolli et al. 2023a MLP, where I don't have access to the code. It may be that some of your claims *a* or *b* may apply to and be true for the Pietropolli et al. 2023a MLP. But then it cannot take hostage of all MLPs, given that it doesn't apply to other MLP models or architectures.)

We have already addressed the comments regarding these two claims in the earlier part of the review when the reviewer first mentioned these sentences (R1).

(c) Could it be that fine scale characteristics of the vertical profile, like interleaved water masses, cannot be well reproduced by the CNN-based PPCon model/architecture? Because it puts more emphasis on the entire/large-scale profile shape and 'neglects' the information from the water mass properties, which is the only information available to MLPs?

We acknowledge this observation and agree that CNNs and MLPs have different strengths.

Our CNN-based approach emphasizes the overall shape of the vertical profile, which is advantageous for predicting variables like chlorophyll and bbp700. These variables benefit from the CNN's ability to learn and reproduce smooth, consistent profiles. For nitrate, which has more regular shape profiles, both CNNs and MLPs show similar performance.

It is important to note that the smoother appearance of our CNN profiles is primarily due to the discretization applied. We used consistent discretization to ensure uniform input and output dimensions for the convolutional architecture. If we were to increase the resolution of discretization, our CNN would likely be able to reproduce fine-scale characteristics more accurately.

We see the CNN approach as complementary to MLPs, rather than superior. Each method has its advantages depending on the specific variable being predicted. For fine-scale details, increasing the discretization for CNNs could enhance their ability to capture these features effectively, similar to MLPs.

(d) Why are the measured data in the manuscript without fine scale and with mostly eroded mixed layer base?

Since CNNs require input vectors of the same length, we used a 5-meter average in the pre-processing step, as explained in (R2). This averaging smooths the profiles, resulting in the loss of fine-scale features and an appearance of an eroded mixed layer base in the measured data. This pre-processing was applied to input data of both MLP and CNN models, keeping the comparative analysis consistent and fair.

**Further points:**

- Figure 1: I cannot find float 6901767, the second nitrate float used later on (Fig. 7 and Tab. 8). Or is it a WMO mix-up there (and it's 6901648??).

We corrected and updated the Figure 1.

[Figure]

- l. 133: "Table 2" referred to is missing.

Thank you for pointing out the missing reference to "Table 2" on line 133. The table in question is located on the previous page of the manuscript and reports the components of the PPCon architecture. We will ensure that the reference is clearly indicated.

- l. 134: From the description of Amadio et al. 2023 (and the above experience) it reads that the float data were treated with the "bit.sea python package (Bolzon et al., 2023)", including "a smoothing task". For a manuscript that focuses and puts emphasis on profile shapes, this is important information that belongs into the description of the dataset (and cannot be hidden somewhere inside some reference). And which warrants some (brief) discussion, because it seems that your figure's mixed layer bases are much more eroded than in the original float data. Which may have implications if PPCon were to be used for augmenting a float dataset.

In the case of the present work, we already explained that we performed a 5-m discretization to have a 200-point length profile for nitrate.
In the case of Amadio paper, the smoothing is done to interpolate the float profiles (either BGC-Argo or reconstructed with MLP) to the layer thickness of the model vertical discretization which varies with depth (about 1.5m at the surface, 5m at a depth of 40m, 10m at a depth of 170m, 25m at a depth of 1000m)
The two tasks respond to different needs.

- section 3.1: The question remains on why such a complex architecture was chosen and what should be gained from 4 separate MLPs per singular input (over replication of input data into a 200x1 vector each) to be fed into the CNN. But I won't insist and just note that this is still weakly motivated.

We thank the reviewer for the comment. We will keep it in consideration for further developments of PPCon.

- l. 182: "adding zero padding to the borders of the input tensor": This requires the input to be normalized, i.e., mean data subtracted (and ideally divided by the standard deviation). A batch normalization is mentioned only for the output tensors in the text, not for the input tensor. From table 2 I understand that batch normalization (BN) is done for every layer's input, is it? So maybe follow the
logic as it is outlined in the table also in this paragraph: Layers consist of BN, SELU and have Dropout.
To compress/decompress information we have kernels/strides, which require padding with …

Thank you for your feedback regarding the normalization and padding of input tensors. You are correct in noting that normalization is essential for ensuring that zero padding is effective.

As outlined in Table 2, batch normalization (BN) is applied to the input of every layer. This includes the input tensor, ensuring that it is normalized by subtracting the mean and ideally dividing by the standard deviation.

- l. 314: "For the nitrate variable, the reconstruction performed by the MLP model is also reported (Pietropolli et al., 2023a)." No, it's not – but it should! :-)
For Figure 3 and also Figure B1, I'd also suggest to use a bit wider spectrum of colour than different shades of blue, to be able to better distinguish the different models.

Regarding the statement on line 314, we intended to convey that the reconstruction performed by the MLP model for the nitrate variable is reported in the appendix.  The sentence will be changed as follows:

"For the nitrate variable, the reconstruction performed by MLP models (Pietropolli et al., 2023a and Fourier et al., 2XXX) is also reported in appendix B"

- l. 317f: "...than the previous MLP architecture by Pietropolli et al. (2023a), but similar to MLPs by Fourrier et al. or Bittig et al."? But my general recommendation would be to drop the second part of the sentence.

Regarding the sentence on line 317, we agree with your recommendation and will drop the second part of the sentence to improve clarity and focus.

- l. 472: That predictions tend to better in deep waters compared to surface waters is true for of all approaches; it's not unique to PPCon. It's a feature due to the different variability of the Ocean's parameter space at depth vs. at the surface, nothing of a particular model's architecture.

We agree with the reviewer's comment. Improved prediction accuracy in deep waters compared to surface waters is a common characteristic of neural network approaches due to the inherent variability in the Ocean's parameter space.

However, we also aim to highlight that our approach, PPCon, demonstrates better performance in deep waters compared to other models. The difference in performance between PPCon and MLP models is more pronounced in deep waters, reflecting the enhanced capability of PPCon in these conditions.

- Fig. B1 caption: 6903153 in ION has no nitrate sensor.

As said in previous response, after double-checking, we noticed that the profile reported does not correspond to the profile contained in the image. We apologize for the mistake in reporting the float profile number. We have updated Figure B1 by changing the float name to the correct one. The correct float cycle is 6901772_145.

**Minor points:**
- l. 42: replace "frequency" by something more fitting? Maybe
- l. 133: "Table 2" referred to is missing.
- l. 145: sample date?
- l. 303: "in the ION[+, SWM, and TYR,] with RMSE values below 0.5 [+unit]"?
- l. 306f.: check "which are the highest ..." – Does this still apply?
- l. 395: "DT" was this abbreviation introduced? Why not spell it out

**Typo's:**
- quite a few instances: "BCG" instead of "BGC"
- l. 41: closing ")"
- l. 263: $\alpha s$ ?
- l. 377: closing ")"
- l. 403: understand
- l. 439; BGC-Argo network
- Fig. B1 caption: 6902904
- l. 102/l. 509: Cross-check the Bittig et al. 2019 reference. That's the one you intended? (technical documentation vs. a published manuscript?)

We appreciate the reviewer's feedback on these points and typographical errors. We have addressed all these issues in the revised manuscript.